



# Present-day geodynamics of the Western Alps: new insights from earthquake mechanisms

Marguerite Mathey[1], Christian Sue[1,2], Colin Pagani[3], Stéphane Baize[4], Andrea Walpersdorf[1], Thomas Bodin[3], Laurent Husson[1], Estelle Hannouz[1], Bertrand Potin[5]

[1] University Grenoble Alpes, University Savoie Mont Blanc, CNRS, IRD, IFSTTAR, ISTerre, Grenoble, 38000, France

[2] Chrono-Environnement Besançon, OSU THETA, University Bourgogne-Franche-Comté, Besançon, 25000, France

[3] Univ Lyon, Université Lyon 1, Ens de Lyon, CNRS, Lyon, 69000, France

[4] IRSN,PSE-ENV/SCAN/BERSSIN, BP 17, Fontenay-aux-roses, F-92262, France

[5] Departamento de Geofisica, Universidad de Chile, Blanco Encalada 2002, Santiago, 8320000, Chile

Correspondence: Marguerite Mathey (marguerite.mathey@univ-grenoble-alpes.fr)

**Abstract**.

Due to the low to moderate seismicity of the European Western Alps, few focal mechanisms are available to this day in this region, and the corresponding current seismic stress and strain fields remain partly elusive. The development of dense seismic networks in the past decades now provides a substantial amount of seismic records down to low magnitudes. The corresponding data, while challenging to handle due to their amount and relative noise, represent a new opportunity to increase the spatial resolution of seismic deformation fields. The aim of this paper is to quantitatively assess the current seismic stress and strain fields within the Western Alps, from a probabilistic standpoint, using new seismotectonic data. The dataset comprises more than 30,000 earthquakes recorded by dense seismic networks since 1989 and more than 2200 focal mechanisms newly computed in a consistent manner. The global distribution of P and T axes plunges confirms a majority of transcurrent focal mechanisms in the overall alpine realm, combined with pure extension localized in the core of the belt. We inverted this new set of focal mechanisms through several strategies, including a seismotectonic



zoning scheme and grid procedure, revealing extensional axes oriented obliquely to the strike

30 of the belt. The Bayesian inversion of this new dataset of focal mechanisms provides a probabilistic continuous map of the style of seismic deformation in the Western Alps. Extension is found clustered, instead of continuous along the backbone of the belt. Compression is robustly retrieved only in the Po plain, which lays at the limit between the Adriatic and Eurasian plates. High frequency spatial variations of the seismic deformation are consistent with surface

35 horizontal GNSS measurements as well as with deep lithospheric structures, thereby providing new elements to understand the current 3D dynamics of the belt. We interpret the ongoing seismotectonic and kinematic regimes as being controlled by the joint effects of far-field forces –imposed by the counterclockwise rotation of Adria with respect to Europe- and of buoyancy forces in the core of the belt, which together explain the high frequency patches of extension

40 and of marginal compression overprinted on an overall transcurrent tectonic regime.



# 1 Introduction

The European Alps are characterized by a complex orogenic history. The alpine belt results from the collision between the African and European plates during the Cenozoic (e.g. Coward and Dietrich (1989); Handy et al. (2010)). Tertiary continental collision followed the late Cretaceous to Eocene closure of the Ligurian Tethys ocean (e.g. Stampfli et al., 1998), leading to the indentation of the European plate by Adria. The anticlockwise rotation of the Adriatic plate with respect to the Europe prevails in the tectonic history of the Western Alps (e.g. D'Agostino et al., 2008; Nocquet and Calais, 2004; Serpelloni et al., 2007). More precisely, multiple collision phases led to nappe stacking as well as folds and thrusts, from the inner zones to the outer front of the belt. The Frontal Penninic Thrust (FPT) is the main compressional structure of Oligocene age along the Western alpine arc (Tricart, 1984). Concurrently to the Miocene progradation of the collision front towards the external zones, the inner part of the belt was affected by orogen-parallel extension, crosscutting earlier compressional structures and resulting in intertwined networks of normal and strike-slip faults (e.g. Champagnac et al. (2006); Sue and Tricart (2003); Bertrand and Sue (2017)). A late phase of orogen-perpendicular extension developed in the core of the Western Alps during the Plio-Pleistocene (Bilau et al., 2020; Sue et al., 2007), leading to the extensional reactivation of the FPT (Sue and Tricart, 1999). This continued tectonic regime is currently revealed by the moderate seismicity of the Western Alps, which is distributed along two main seismic arcs, namely the Briançonnais seismic arc running along the FPT, and the Piémontais seismic arc running along the western side of the Ivrea Body (Sue et al., 2002; Figure 1). Congruent with the late brittle extension, ongoing seismicity shows normal faulting combined with dextral longitudinal strike-slip (Sue et al., 1999, 2007; Delacou et al., 2004).

From the end of the 90's onward, seismotectonic studies brought increasing constraints on the current strain and stress states of the Western Alpine realm as a whole (e.g. Maurer et al 1997; Kastrup et al., 2004; Eva et al., 2020). The progressive development of dense permanent and temporary seismic networks in the Western Alps (e.g. Sismalp (Thouvenot et al., 1990, 2013), CIFALPS (Zhao et al., 2013)) improved the catalogues by decreasing both the magnitude of completeness and the minimum magnitude from which focal mechanisms could be computed. These updated catalogues opportunely permit to reappraise the disputed current dynamics of the Western Alps (e.g. Nocquet et al., 2016; Sternai et al., 2019; Champagnac et al., 2012; Mazzotti et al., 2020).



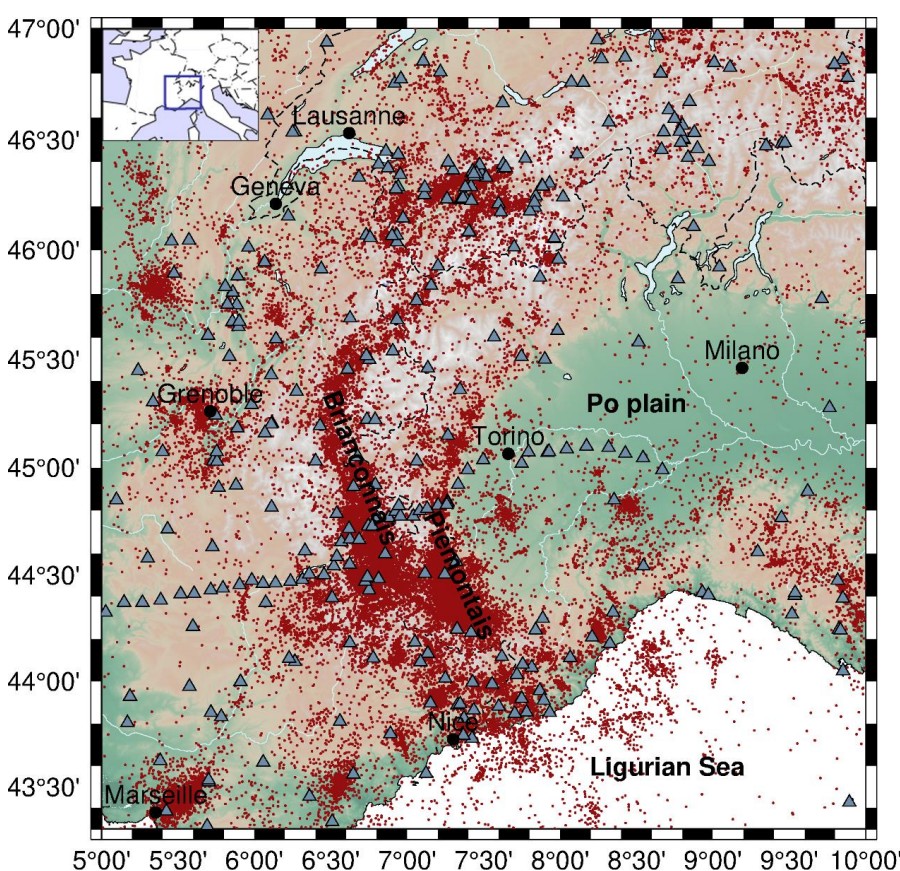

Figure 1. Map of the seismicity (red dots) encompassing the western Alps, compiled in Potin (2016) from national and regional networks seismic stations (triangles). The two major seismic arcs (Briançonnais and Piémontais arcs) stand out clearly from the seismicity. Quarry blasts have not been removed.

In this paper, we apply an integrated seismotectonic approach taking advantage of a comprehensive seismic dataset (Potin, 2016) that compiles 25 years of seismicity recording by national and regional networks and includes more than 30,000 events with local magnitudes (Ml) in the 0 to 5 range. Given the exceptional amount of data available in this catalogue we unveil the highest resolution stress and strain fields to date based on the computation of more than 2200 focal mechanisms, and we retrieve the robust deformation signals at the regional scale. This is achieved thanks to a Bayesian inversion for the deformation style, combined to a stress-oriented inversion of the focal mechanisms. Principal stress orientations, combined with the regionalization of deformation modes, are of primary importance to understand the seismogenic processes within the belt and to seismic hazard analyses.



## 2 Data and methods

### 2.1 Seismic database and earthquake localizations

Our large dataset is a compilation of P and S wave travel time arrivals recorded by six local or national networks operated from 1989 to 2014 (Potin, 2016, Figure 1). (i) The Sismalp network from Grenoble University (Thouvenot et al., 1990, 2013; Thouvenot and Fréchet, 2006) consisted of up to 44 permanent and 10 temporary stations, from 1989 and 2013, and specifically targeted the weak seismicity of the Western Alps. (ii) The French RéNaSS (*Réseau National de Surveillance Sismique*) and LDG (*Laboratoire de Détection Géophysique*) networks were used to build the BCSF catalogue. These networks entirely cover France, and include 36 stations in the Western Alps. (iii) The Italian RSNI network (Regional Seismic Network of North Western Italy) comprises 40 stations in the Western Alps and the Po plain and collects data since the 1960s onward. (iv) The Swiss SED network (Cauzzi and Clinton, 2013) maintains an increasing number of stations in Switzerland since the 1970s onward with about one hundred stations as of today. (v) The CIFALPS network (Zhao et al., 2013) recorded between 2012 and 2013 along a profile across the SW Alps. Potin (2016) standardized and homogenized the seismic data collected from these different networks: duplicate events were removed, and arrival time uncertainties were harmonized. Potential picking errors were identified and cleared out. The remaining dataset includes 36,010 events, for which at least seven phases were read by at least four seismic stations. 791,754 P- and S-waves arrival times were retained, recorded at 375 stations. Potin (2016) combined 3D-velocity model inversion and earthquake re-localizations; take-off angles of seismic rays were estimated with an uncertainty of the order of a few degrees, thus providing an enhanced localization of the events with a precision of a few kilometers both laterally and vertically. The complete set of earthquakes includes blasts, quarrying or mining events.

### 2.2 Focal mechanism computation

From the aforementioned large dataset, we computed focal mechanisms with the code HASH (Hardebeck and Shearer, 2002). The computation makes use only of P-wave first motion polarities. Thanks to the high density of stations provided by the combination of six networks, we were able to apply strict computation criteria. From the 36,010 events of the above described



dataset, we retained those with at least 10 P-wave polarities and at least one S-wave. The
maximum allowed azimuthal gap between polarities was set to 90° and the maximum azimuthal
gap of incidence angles to 60°. The maximum allowed distance to the seismic station was set
to 600 km since all the stations used are within this range and covered by the same velocity
model from Potin (2016). A preferred focal mechanism was computed for each of the 2,215
events meeting these criteria. HASH code yields a quality flag from A to F for each computed
focal mechanism (A, best constrained), which takes into account several parameters such as the
number of polarities used, spatial distribution of stations, uncertainties on picking and take-off
angles. The local magnitude (Ml) is based on the maximum amplitude of the shear-wave. The
quality flags distribution of the computed focal mechanisms is available in Figure S1. Over the
2,215 events for which a focal mechanism has been computed, 58 have a Ml<1; 1,200 have a
Ml ranging from 1 to 2; 769 have a Ml ranging from 2 to 3; 102 have a Ml ranging from 3 to 4,
and 19 have a Ml>4 (Figure 2a). 15 of them are A quality events with Ml ranging from 1.6 to
4.6; 52 are B quality events with Ml from 1.3 to 4.9; 72 are C quality events with Ml from 0.9
to 4.8; all other mechanisms are D quality events with Ml ranging from 0.2 to 4.5. According
to Hardebeck and Shearer (2002), A quality events have an associated fault strike uncertainty
between 0° and 25°, B events between 25° and 35°, C events between 35° and 45°, and D events
between 45° and 55°.

While low magnitude seismic events account for less seismic energy release compared to larger
ones, they will nonetheless be used both in an approach in which all focal mechanisms are used
regardless of their magnitude (stress inversions, section 2.3), and in approaches in which focal
mechanisms are weighted depending on their magnitude (seismic moment summation, section
2.3) and overall quality (Bayesian inversion, section 2.4), respectively.

To further analyze the regional distribution of the focal mechanisms, we classified them
according to the plunges of their pressure (P), tension (T) and null (B) axes (following the
method of Frohlich (1992)). The representation of the style of deformation of a given focal
mechanism between pure strike-slip, pure normal or pure reverse motion is a problem where
the mechanism is located in a ternary diagram. Each pole of the diagram represents one of the
three pure styles of deformation (pure strike-slip, pure normal or pure reverse motion). For each
focal mechanism, Kaverina diagrams (e.g., Alvarez-Gómez, 2019) allow to represent the styles
of deformation including the case of intermediate modes such as transpression or transtension.
Frohlich's classification with Kaverina diagrams allows to represent a focal mechanism in the




ternary diagram by only two coordinates, the B and T axes plunges, P axis plunge being retrieved by the product of the 2 first ones (*e.g.* if both T plunge and B plunge = 0 °, then P plunge = 90 °). The representation of the style of deformation through the Frohlich classification thus allows one to project a focal mechanism into a 2D parameter space, and to

account for intermediate styles of deformation, while information on the plane orientation is lost. The Kaverina diagram for the complete catalogue is presented on Figure 2b, and the corresponding map on Figure 2c.

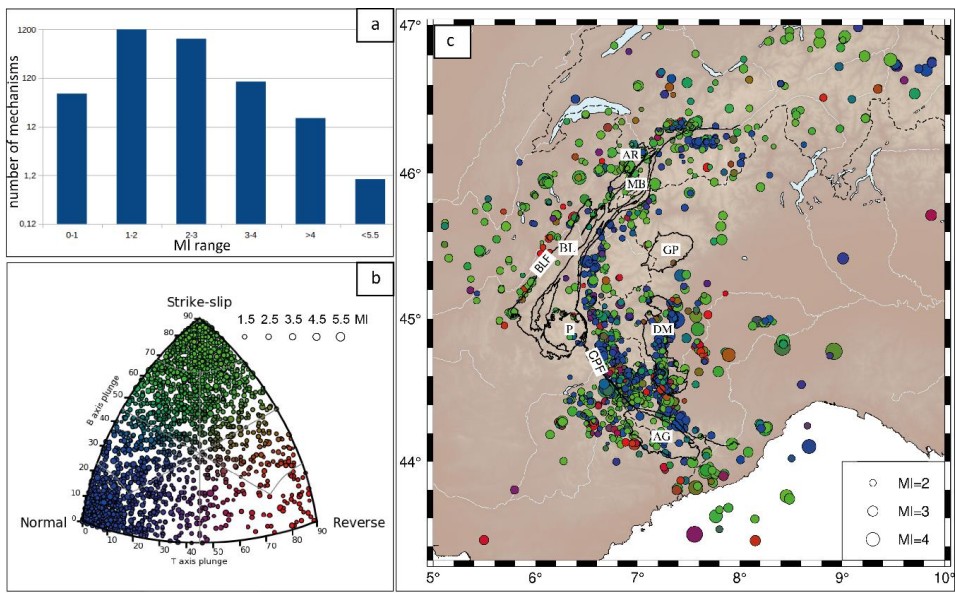

Figure 2. Set of the 2215 focal mechanisms computed in this study. a) Histogram of magnitude
distribution (local magnitude, Ml) for the computed focal mechanisms. b) Kaverina diagram displaying the style of deformation of the mechanisms according to their T axis (tension axis) and B axis (null axis) plunges. Normal, reverse and strike-slip deformation style corresponding to vertical P, T and B axes is indicated by circles in blue, red and green color, respectively. c) Localization of the focal mechanisms color-coded according to their classification in the
Kaverina diagram in b). Note the different scale for events size between b) and c). Black lines outline the crystalline massifs of the area. External crystalline massifs (Aiguilles Rouges, AR; Mont-Blanc, MB; Belledonne, BL; Pelvoux, P; Argentera, AG) are separated from the internal crystalline massifs (Gran Paradison, GP; Dora Maira, DM) by the Crustal Penninic Front (CPF). The Belledonne external massif is delineated by the Belledonne Fault (BLF) along its western
flank.



### 2.3 Strain quantification and stress inversions

Strain rates are computed by averaging moment tensors (*i.e.*, symmetrical 9 components 2nd
order tensor, plus seismic moment amplitude), for which the 9 components directly depend on
strike, dip and rake parameters of the focal mechanisms. The seismic moment for each focal
mechanism is by definition related to moment magnitude Mw (Hanks and Kanamori, 1979).
Our Sismalp catalogue includes a local magnitude scale (Ml) for which a double conversion
has been proposed by Cara et al. (2015), through another national-scale local magnitude (Ml
LDG). However, a careful analysis of this double conversion (Laurendeau et al., 2019) shows
that this relationship is misleading for Ml Sismalp >2.5: above this value, a linear relationship
could be derived from the available dataset; below this value, a polynomial equation is a better
approximation. Finally, Mw=Ml is a reasonable hypothesis, however under-estimating moment
magnitude on the complete range of values. In order to retrieve an annual strain rate, the sum
of the individual moment tensors is divided by a volume and a time span, according to Kostrov
(1974). This method thus requires defining homogeneous volumes in terms of deformation
style, prior to summing compatible focal mechanisms in each volume. We defined 11 volumes
within our region of interest, based on structural criteria, taking into account both the arcuate
structure of the belt, as well as its geological structures, or based on the focal mechanism
distribution and density of earthquakes. The 11 seismotectonic zones are named according to
their structural environment: the VSS and VSN zones in the Southern and Northern Valais;
BRN, BRS, BRS2 from North to South along the Briançonnais seismic arc; PIE along the
Piémontais seismic arc; PPO, SMT, NMT, and VAR for the areas at the periphery of the belt:
Po plain to the East, Southern Mercantour to the south, and Northern Mercantour and Var to
the Southwest, respectively; and finally DPH in the external Dauphinois zone (map view on
Figure 3).

In a second analysis, focal mechanisms were inverted to retrieve the principal stress directions.
Different inversion methodologies exist, all based on the same assumption of a locally uniform
stress regime in the crust producing a slip, which occurs in a direction parallel to the plane of
resolved shear stress (Hardebeck and Hauksson, 2001; Lund and Townend, 2007). In the stress
inversion strategy, all mechanisms are equally weighted, regardless of their magnitude. In order
to strengthen our analysis, we compare the classic FMSI (Focal Mechanism Stress Inversion,
Gephart, 1990) method based on a grid search algorithm with the SI (StressInverse, Vavryčuk,
2014) method based on a linear least-square inversion. These methods allow retrieving a partial



stress tensor, since orientations of the three principle stresses can be obtained, as well as the shape ratio of the stress tensor, but not their absolute amplitude. The retrieved information is constitutive of the deviatoric stress tensor (D'Amico, 2018). In a first step, these two procedures were implemented on the same seismotectonic zones that were defined for strain rate quantification, in order to assess the robustness of the inversion. In a second step, we inverted

the mechanisms on a regular grid rather than on pre-defined seismotectonic zones, which enables us to increase the spatial resolution of the derived stress field, at the cost of a reduced level of constraint since less data are used to derive each tensor. We used the MSATSI (Matlab Spatial And Temporal Stress Inversion, Martínez-Garzón et al., 2014) software, which is based on the SI algorithm (Vavryčuk, 2014) to perform an inversion for each cell encompassing at

least 10 focal mechanisms. While MSATSI allows the user to modulate the damping factor describing the attenuation of the weight of neighboring cells in a given cell inversion, we choose to run the inversion without damping in order to identify any spatial heterogeneity in the stress field.

**2.4 Bayesian interpolation of focal mechanisms**

We choose to interpolate the style of deformation, in order to construct a continuous map of the regional trends prevailing among the locally varying mechanisms. As shown in Figure 2, our dataset consists in an ensemble of P and T axes plunge angles given at each event location. In order to better investigate the spatial variations of this dataset, we implemented a 2D Bayesian regression method (after Bodin et al., 2012) to reconstruct a continuous surface of P and T

plunge angles. The procedure is based on a transdimensional regression, which can be used over n-dimensions datasets that are evenly distributed, and of variable uncertainties. The reconstructed surface is parameterized with a mesh that self-adapts to the level of information in the data. The solution is a full probability distribution for each parameter at each geographical location, which is useful to estimate uncertainties. This method was first use to reconstruct the

Moho topography beneath Australia from a discrete set of local observations (Bodin et al., 2012). Choblet et al. (2014) used it to reconstruct probabilistic maps of coastal relative sea level variation from tide gauges records. The approach was also used by Husson et al. (2018) to reconstruct maps of vertical displacement rates measured at GPS stations, and by Hawkins et al. (2019a;b) to produce probabilistic maps of sea level rise by combining GPS, satellite

altimetry and tide gauge measurements.



In this work, the transdimensional regression algorithm applied to focal mechanisms outputs probability distributions for P and T plunges values at each geographical location. The method accounts for the heterogeneous data density thanks to a self-adapting parameterization based on Delaunay triangles. It also accounts for variable uncertainties in the data and thus allows

deciphering which signal is robust at the regional scale despite the data heterogeneity. Moreover, a hierarchical Bayesian approach is used and data uncertainties are re-scaled by a global adaptive factor depending on the level of data fit (Malinverno et al., 2004). This factor enables us to assess whether formal uncertainties on fault planes and thus on P and T axes are over- or underestimated.

**3 Results**

**3.1 Focal mechanism distribution**

To get a general overview of the 2200 focal mechanisms in terms of mode of deformation, we plotted them all together on a Kaverina diagram, *i.e.* according to their B and T axes plunges. The corresponding plot (Figure 2b) shows a majority of strike-slip mechanisms (~ 1200; 54.5

%), a large number of extensional ones (~ 800; 36.5 %), and a minority of reverse ones (~ 200; 9%), over the Western Alps as a whole. The localization of the events is shown on the map in Figure 2c, with respect to the main geological features covering the area. At a first glance, a strike-slip regime is distributed all over the belt, while extension is mainly located in the inner part of the belt, *i.e.* along the Briançonnais and Piémontais arcs, confirming previous studies

(Bauve et al., 2014; Delacou et al., 2004; Sue et al., 2002). Compression is scattered, and mostly expressed in the Po plain and in the S-W external part of the belt. In terms of tectonic structures, the Belledonne fault (BLF) is well imaged by the focal mechanisms aligned over ~100 km along the external flank of the Belledonne crystalline massif. This active structure is dextral (e.g. Martinod et al. (1996) Billant et al. (2015)) and accounts for a large part of the seismicity

released in this area during the instrumental period (Thouvenot et al., 2003). At a larger scale, a majority of the events are indeed located within the Briançonnais or Piémontais seismic arcs, which are characterized by complex parallel fault networks. Within these arcs, some surface fault traces are mapped from field works and could be associated with focal mechanisms at the local scale (for instance the High Durance and East-Briançonnais Faults (Mathey et al., 2020;

Sue and Tricart, 2003)). However, due to the large spatial scale of the present study, we did not perform any detailed analysis of focal mechanisms distribution with regard to individual faults.





In a second step, we plotted Frohlich classes on Kaverina diagrams in each of the 11 predefined
seismotectonic zones (Figure 3). These plots confirm that the 11 zones display roughly
consistent deformation modes; the homogeneity of the deformation mode in each zone being a
prerequisite to strain estimates. Importantly, strike-slip mode prevails everywhere, either as
almost pure transcurrent zone in the Dauphinois and northern Valais (DPH, VSN), or in
association with extension along the Briançonnais and Piémontais seismic arcs (BRS, BRS2,
BRN, PIE, SMT). The reverse fault end-member of the diagram is prevailing in the Po plain
(PPO) zone only, but also appears noticeably in the Piémontais and Var zones (PIE and VAR).

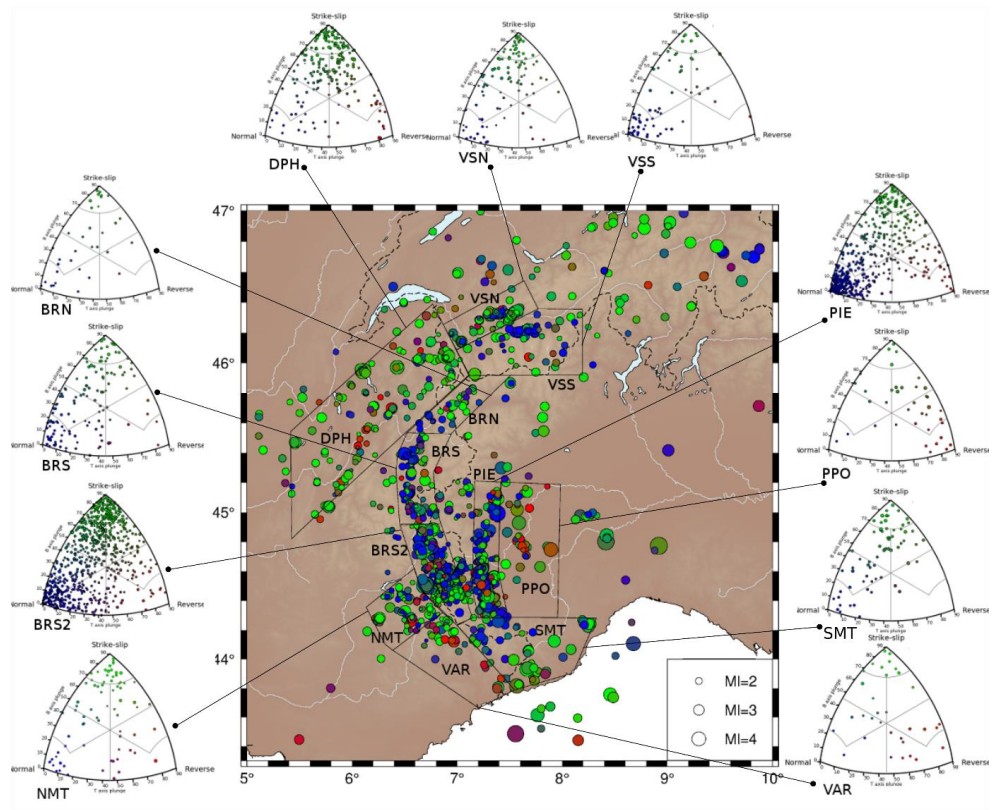


Figure 3. Seismotectonic zonation of the study area yielding 11 zones based on structural
criteria, style of seismic deformation and its density. VSS: Valais South; VSN: Valais North;
BRN: Briançon North; BRS: Briançon South; BRS2: Briançon South 2; PIE: Piemontais
seismic arc; PPO: Po plain; VAR: Var; NMT: Northern Mercantour; SMT: Southern
Mercantour; DPH: Dauphinois zone. Kaverina diagrams are shown for each seismotectonic
zone, highlighting their homogeneous style of deformation, associating either normal and
strike-slip events or reverse and strike-slip events.



### 3.2 Up-to-date strain and stress fields

The strain rate tensors computed in each of the 11 seismotectonic zones with Kostrov's (1974)
method are presented projected at the surface in Figure 4 and the related parameters (rate,
azimuth and dip) are listed in Table 1. The largest strain tensor of the belt is located in the
southern Briançonnais area (BRS2), where the extension axis reaches 0.3 nanostrain/yr trending
WNW-ESE, and the compressive one 0.27 nanostrain/yr trending NNE-SSW. The PIE and

DPH zones also present noticeable strain tensors. For PIE, extension is evaluated to 0.09
nanostrain/yr oriented N-S (and compression to 0.1 nanostrain/yr), and for DPH, extension and
shortening of similar magnitude (0.05 nanostrain/yr each) are oriented NNE-SSW and WNW-
ESE respectively. The deformation style of the strain tensors is generally strike-slip (both
extension and compression horizontal), except for Piémontais (PIE) and southern Mercantour

(SMT) areas, which show vertical compression axes and negligible compression rates
respectively. However, in the Kostrov methodology, the style of deformation and its orientation
are dominated by the larger events in each sub-area, as they bring most of the energy released.
The short time span of the observations (25 years) prevents from properly investigating the
time/energy relation per zone. Indeed, including in the summation the historical events that

occurred in the Western Alps could modify the strain rate distribution in the belt.

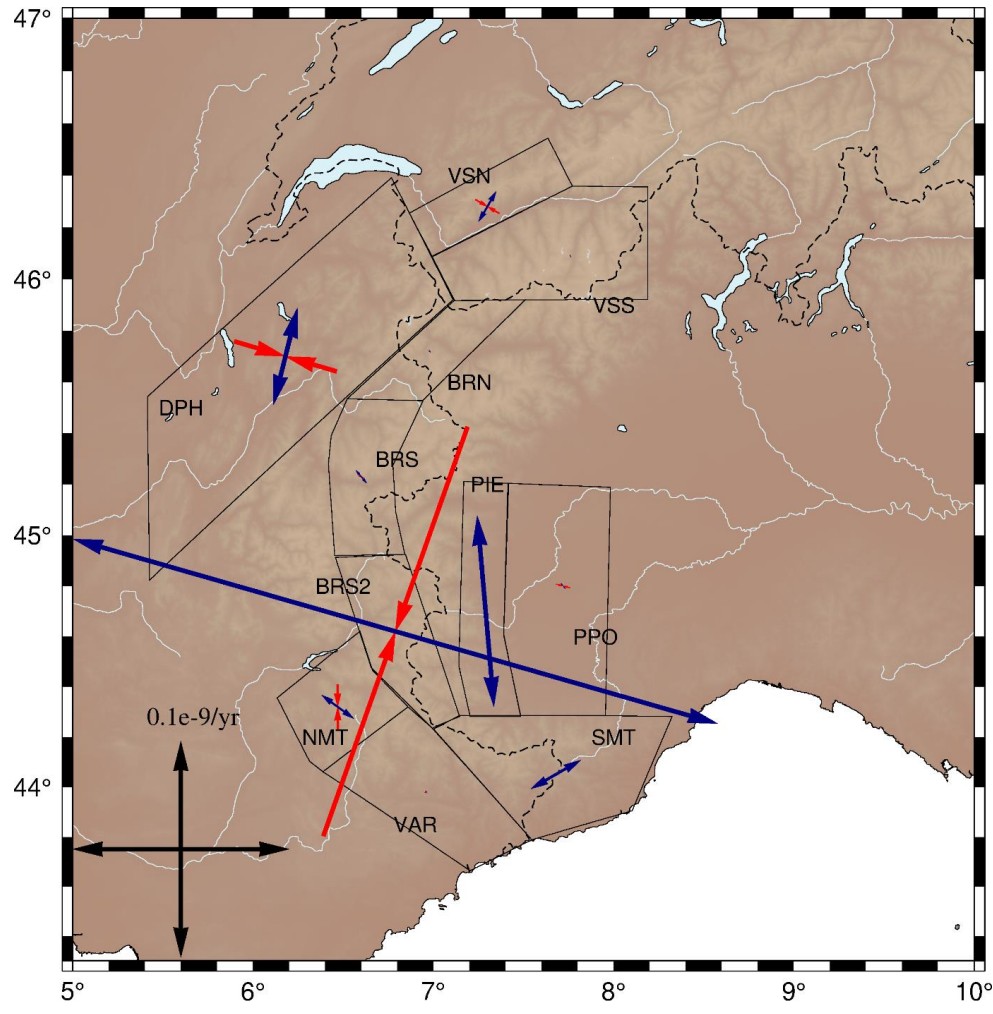

Figure 4. Seismic strain rates computed according to Kostrov's (1974) method in each of the 11 subareas, based on the 25 years of seismicity records from Potin (2016). Blue and red arrows show respectively the extensive and compressive components of the strain rate projected at the surface ($\dot{\varepsilon}*\cos(\delta)$), and black arrows represent the scale of the strain rate. Values used for projection are reported in Table 1.







| | Kostrov strain rates | |
|---|---|---|
| | $\dot{\varepsilon}1$ (yr⁻¹) ; θ (°) ; δ (°) | $\dot{\varepsilon}3$ (yr⁻¹) ; θ (°) ; δ (°) |
| BRN | -9.8e⁻¹³ ; -124 ; 7 | 1.2e⁻¹² ; -33 ; 7 |
| BRS | -8.4e⁻¹² ; -69 ; 65 | 7.9e⁻¹² ; -41 ; 9 |
| BRS2 | -2.75e⁻¹⁰ ; 19 ; 42 | 3.1e⁻¹⁰ ; 107 ; 3 |
| DPH | -4.9e⁻¹¹ ; -74 ; 6 | 4.7e⁻¹¹ ; 14 ; 13 |
| NMT | -2.1e⁻¹¹ ; 179 ; 11 | 1. 8e⁻¹¹ ; 126 ; 71 |
| PIE | -1.1e⁻¹⁰ ; 121 ; 89 | 8.9e⁻¹¹ ; -5 ; 0 |
| PPO | -6.9e⁻¹² ; 105 ; 20 | 7.9e⁻¹² ; -37 ; 65 |
| SMT | -8.4e⁻¹⁴ ; 39 ; 69 | 2.9e⁻¹¹ ; -121 ; 20 |
| VAR | -1.8e⁻¹² ; -39 ; 18 | 1.2e⁻¹² ; 64 ; 36 |
| VSN | -1.6e⁻¹¹ ; 120 ; 36 | 1.6e⁻¹¹ ; -149 ; 0 |
| VSS | -7.3e⁻¹³ ; 26 ; 57 | 1.3e⁻¹² ; 148 ; 19 |

Table 1. Strain tensors (Kostrov's method) for each of the eleven surbarea. Strain rates display first ($\dot{\varepsilon}1$) and third ($\dot{\varepsilon}3$) eigenvalues of the strain tensor corresponding to compression and extension respectively, as well as their azimuth (θ, in degrees from north) and dip (δ, degrees) used to project the strain tensors at the surface in Figure 4.

To better explore the current tectonic regime in each sub-area, we investigated the distribution of stress orientations using focal mechanism inversions. All inverted earthquakes are equally weighted, regardless of their magnitude. Figure 5 shows the azimuths of maximum (σ1) and minimum (σ3) compressive stresses, as given by SI and a FSMI inversions, plotted according to their plunge (*i.e.* the length of the stress axes being expressed as a unitary

value*cos(plunge)). The three principle stresses being orthogonal, horizontal σ1 alone (*i.e.* σ1 axis length at the maximum unitary value) represents a compressional regime, horizontal σ3 alone (*i.e.* σ3 axis length at the maximum unitary value) gives an extensional regime, and both horizontal σ1 and σ3 (*i.e.* both σ1 and σ3 axis lengths at the maximum unitary value) indicate a strike-slip regime. The most representative focal mechanism in each zone is plotted beside

each sub-area. Associated azimuth and dip values as well as strike, dip, and rake of both inversions and the corresponding mean mechanism from SI inversion are listed in table 2. The FMSI inversion additionally reports a misfit value, listed after the principle stress components.





The results of the two inversion methods are in rather good agreement for each of the 11 zones, highlighting the overall stability and the robustness of our stress inversions.

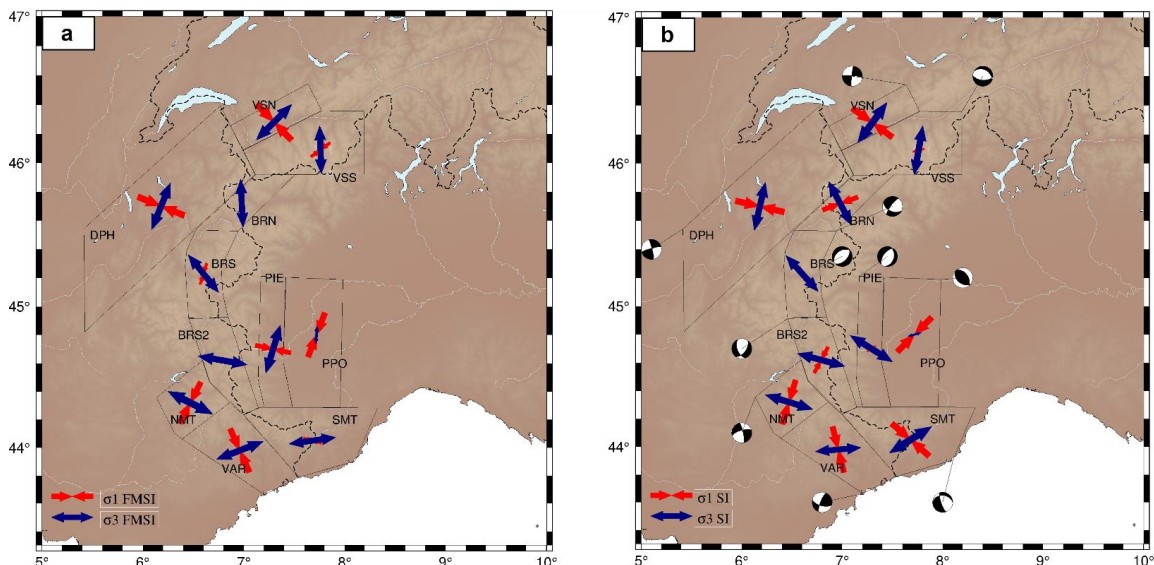

Figure 5. Comparison of stress orientations resulting from FMSI (a) and SI (b) inversions of focal mechanisms within each subarea. Most compressional (σ1, in red) and least compressional (σ3 in blue) stresses are plotted given their plunge (*i.e.* arrows length depending on the plunge: the scale arrows in the bottom left corner represent 0° plunging stress axes (horizontal stresses), while 90° plunging stress axes will not be seen on the map). The mean

focal mechanism retrieved from SI inversion, and corresponding to each partial stress tensor, is plotted for each subarea in b).

| | FMSI stress orientations | | | | SI stress orientations | | | mean mechanism | | |
|------|------|------|------|-------|------|------|------|------|-----|------|
| | σ1 (θ;δ) | σ2 (θ;δ) | σ3 (θ;δ) | misfit | σ1 (θ;δ) | σ2 (θ;δ) | σ3 (θ;δ) | strike | dip | rake |
| BRN | 173;73 | 267;1 | 357;17 | 6.242 | 70;41 | 232;46 | 332;8 | 217 | 76 | -138 |
| BRS | 197;64 | 57;20 | 321;15 | 10.842 | 160;73 | 50;5 | 318;15 | 45 | 50 | -97 |
| BRS2 | 135;80 | 11;6 | 280;8 | 10.195 | 28;54 | 188;33 | 285;9 | 36 | 55 | -47 |
| DPH | 292;1 | 188;86 | 22;4 | 8.710 | 283;1 | 21;80 | 193;9 | 255 | 82 | 174 |
| NMT | 207;23 | 26;67 | 117;0 | 8.088 | 197;23 | 14;66 | 107;1 | 348 | 79 | -159 |
| PIE | 282;44 | 112;46 | 17;5 | 21.324 | 238;78 | 31;9 | 122;5 | 217 | 57 | -78 |
| PPO | 21;20 | 289;5 | 184;69 | 11.130 | 45;13 | 137;7 | 254;74 | 323 | 47 | 99 |
| SMT | 106;68 | 355;8 | 262;20 | 9.556 | 131;58 | 334;29 | 238;10 | 161 | 77 | -60 |

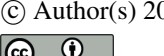



| | | | | | | | | | | |
|---|---|---|---|---|---|---|---|---|---|---|
| VAR | 336;19 | 176;70 | 69;6 | 9.737 | 346;19 | 223;57 | 85;24 | 202 | 77 | 30 |
| VSN | 136;2 | 268;86 | 45;3 | 7.397 | 126;6 | 312;83 | 216;0 | 276 | 86 | -175 |
| VSS | 234;62 | 96;22 | 358;17 | 7.905 | 256;76 | 101;11 | 10;5 | 94 | 58 | -104 |

Table 2. Stress (FMSI and SI) results for each of the eleven subareas. Stress orientations display the azimuth (θ, in degrees from north) and the dip (δ, degrees) of the most compressional stress (σ1), intermediate stress (σ2), and least compressional stress (σ3) for the two methods. FMSI also yields a misfit value (dimensionless) for the deviatoric stress tensor constituted by the three principal stresses. SI yields the focal mechanism corresponding to the inverted deviatoric stress tensor.

The extension axis (σ3) presents close azimuth values between the two solutions, excepted for Piémontais sub-area (PIE), with a difference of 105° between σ3 azimuths derived from SI and FMSI inversions, and for the Po plain (PPO) zone which shows a 70° σ3 azimuth difference while both methods highlights a dominant compressive tensor in this area (sub-vertical σ3 axis associated to sub-horizontal σ1 axis). Some azimuth and plunge differences are also observed for the compression axis (σ1) between the two solutions, but mainly for areas in which extension prevails (σ1 sub-vertical), such as in the BRN and BRS2 areas. Our results thus specifically suggest that the zoning realized for the Piémontais arc mixes up seismicity from different tectonic regimes, either belonging to different fault segments or different depths. In the overall the set of inversions allows deciphering six mostly extensive sub-areas in the core of the belt (VSS, BRN, BRS, BRS2, SMT, PIE), four strike-slip sub-areas at the periphery (VSN, DPH, NMT and VAR) and only one compressive sub-area in the Po Plain (PPO).

To test the significance of the observed heterogeneity between the inner and outer parts of the belt, we implemented a MSATSI stress inversion (based on SI method, see above) on a 0.5°x0.5° geographical grid for each cell encompassing at least ten focal mechanisms (Hardebeck and Michael, 2006). This inversion procedure allows to get past any a priori seismotectonic zoning for the Alpine stress determination along the arc of the orogen. Moreover, it increases the spatial resolution of the derived stress field, which may enable to account for a varying stress field at higher spatial frequency such as in the Po plain. Such an inversion on a regular 0.5°x0.5° grid (Figure 6) is possible for the first time in the Alps thanks to the large number of focal mechanisms computed in our dataset, resulting in the highest resolution stress map to date in the Alps. It is worth noting that neighboring stress tensors appear consistent in orientation. The orientations are also in good agreement with the stress tensors derived in the seismotectonic zoning scheme.



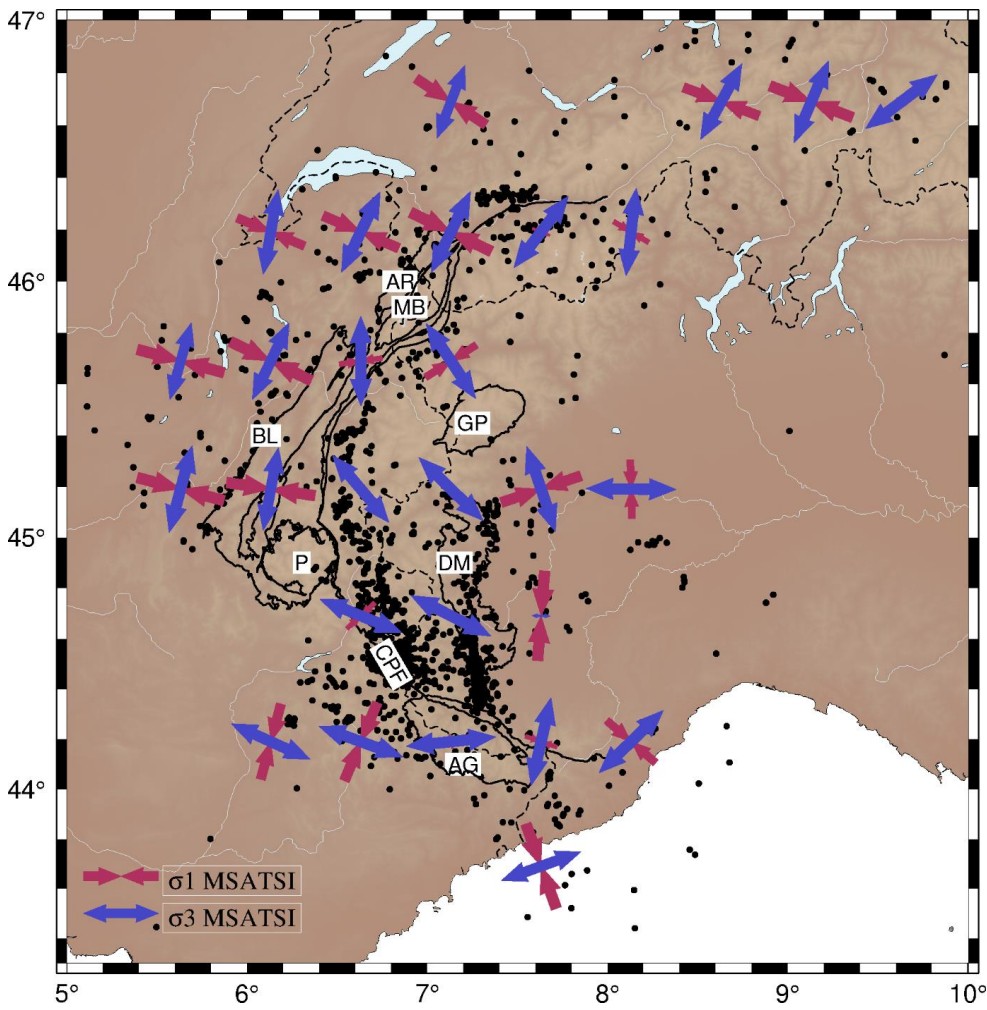

Figure 6. Stress orientations resulting from MSATSI inversion (based on the SI method) on a 0.5°x0.5° grid, for all cells containing at least 10 focal mechanisms (black dots). Most compressional (σ1, in red) and least compressional (σ3, in blue) stresses are projected according to their plunge. Solid lines and their labels refer to major geological features described Figure 2.

Extension is localized along the Briançonnais and Piémontais seismic arcs (Figure 1) as previously shown (review in Sue et al. (2007b); section 4), but it is found oblique to the strike of the Alpine belt, whereas it has been often described as perpendicular to the orogen (Sue et al., 2007b). The angle between the direction of extension and the strike of the belt is of the order of 30° to 40°. This means that the extensional direction is systematically deflected clockwise with respect to the direction normal to the arc. This feature is observed all along the arcuate



shape of the belt, from the Valais area in the North to the southernmost tip of the Alps in the South. All the western periphery of the belt (corresponding to the zones VSN, DPH, NMT, VAR Figure 5) show strike-slip stress fields, with a rotating state of stress compatible with dextral motions along longitudinal directions (typically along longitudinal faults such as the Belledonne fault, Thouvenot et al., 2003). Compression is once again only retrieved in the cell centered on the Po plain, with an axis oriented almost N-S. This grid-inversion shows the prevalence of strike-slip around the bend of the Western Alps, as well as the deflection of the extension in the core of the orogen with respect to its strike, and the very limited importance of compression.

**3.3 Probabilistic reconstruction of tectonic regimes throughout the Western Alps**

Spatial variations of the deformation modes are assessed though the Bayesian reconstruction of continuous surfaces for P and T axes plunges. This is achieved through a 2D regression, where hypocenter locations are first projected at the surface and depth is ignored. From a discrete set of values (P and T axes plunges for each event) distributed on a 2D map, the problem consists on reconstructing a smooth surface for these values. As shown with the earthquake classification in section 3.1, these two parameters are sufficient to describe the style of deformation. The Bayesian approach provides a full probability distributions for the P and T axes at any point in a 300x300 km² area covering the entire western part of the belt. From these two distributions, a continuous probabilistic map of the deformation style can be constructed in the region.

The information on the style of deformation retrieved from the Bayesian inversion, once combined to the information of the seismic energy released in the area, thus delivers a comprehensive view of the current seismic deformation in the western Alps. Figure 7a shows a smoothed map of the seismic flux in the area. This map was generated by interpolating the seismic moment (Mo) values of events in the full dataset described in section 2.1. The seismic moment is estimated assuming that Ml=Mw in our region (section 2.3) and using Hanks and Kanamori (1979) magnitude-moment relationship. Mo values are then summed on the 25 years long period (the raw seismic moment map is displayed in Figure S2) and annualized. It appears from Figure 7a that most of the seismic energy was released during the considered time interval along the Briançonnais arc, from the Briançon region to the Valais region, and along the Piémontais arc.



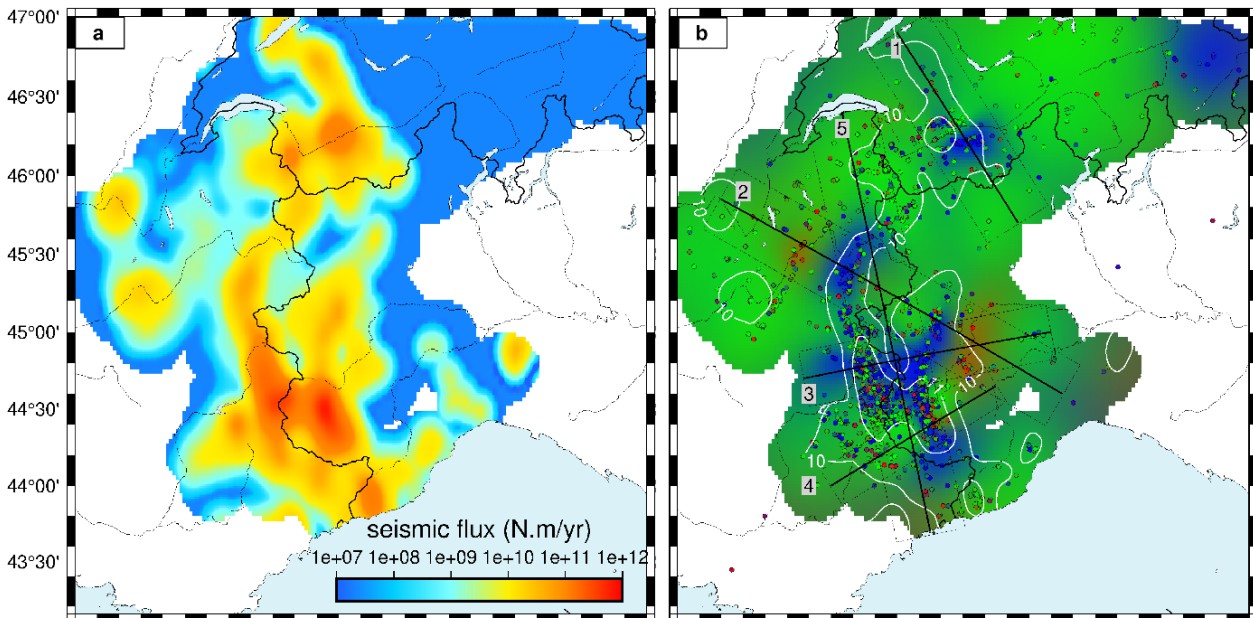

Figure 7. a) Seismic flux computed using the annualized seismic moment (Mo) over the 1989-
2014 time interval, summed in 0.05°x0.05° cells and interpolated with an adjustable tension
curvature surface algorithm (Smith and Wessel, 1990). b) Mean values of the distributions of
P, T, and B axes plunges resulting from a Bayesian inversion. The color-code is according to
the Kaverina classification of the style of deformation (the end members red, blue and green
corresponding to reverse, normal and strike-slip deformation style, respectively). Focal
mechanisms used are represented as dots filled with the same color-code. Areas without focal
mechanisms within 35 km are masked out, as well as mechanisms in the Ligurian sea, due to
the lack of constraints. White lines show isocontours of the seismic flux from a) in Log10(Mo).
Profiles numbered from 1 to 5 correspond to the interpolated vertical cross-sections in Figure
9. Dashed boxes encompass the focal mechanisms projected along each profile on Figure 9.

        Figure 7b shows a map of deformation modes obtained by combining the mean values

of the Bayesian posterior distributions for P and T axes plunges, converted to a color code in

r/g/b triplet at each geographical location, consistently with the ternary diagram representation

(Figure 2b). Transcurrent and transtensive deformation predominate all over the belt. Extension

is specifically localized in the core of the belt, as shown by previous studies (review in Sue et

al. (2007b)). However, whereas previous works have shown a continuous stretch of extension,

our interpolation shows much more localized extensional areas. They are located

discontinuously along the Briançonnais seismic arc (running along the CPF from the southern

Briançonnais region to the Swiss Valais region, Figure 1) and the Piémontais seismic arc and

embedded within an overall strike-slip regime that prevails throughout (Figure 7). The

extensional areas correspond in the overall to areas of higher seismic energy release, even if





some strike-slip areas also bear significant energy such as in the Aiguilles Rouges/Mont-blanc

crystalline massifs (Figure 6). Compressive patterns are retrieved from our interpolation in the

Po plain, as seen above in the stress inversion, but also in a very small area of the external zone

along the Belledonne fault. Figure 8 shows the standard deviations of the P and T axes plunges

probability distributions (see section 2.2). Uncertainties range between 5 and 20 degrees on

interpolated P and T axes plunges. Transitional areas between pure styles of deformation, and

areas at the border of the network are less constrained as shown by higher uncertainties. Figure

8b shows that on the two compressive patterns retrieved, only the one lying east of the

Piémontais arc appears as a robust feature.

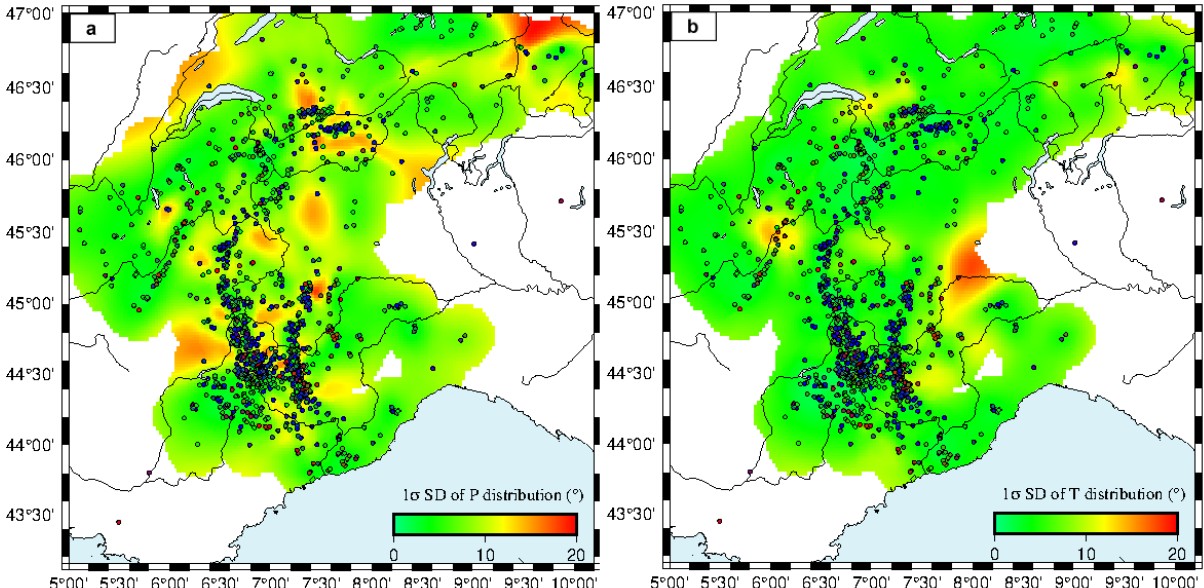

Figure 8. 1-sigma uncertainties (in degrees) on the mean values of P distributions (a) and of T distributions (b) resulting from the Bayesian interpolation in Figure 7. Superimposed focal mechanisms allow us to distinguish between two types of higher uncertainty areas: areas

without enough data, and areas displaying high data heterogeneity.

To investigate the spatial variability of deformation mode at depth, and thus obtain a

3D-view of the overall deformation in the belt, we perform in a second step 2D inversions over

five vertical cross-sections across the belt (Figures 7 and 9). To do so, we first projected focal

mechanisms on the 2D cross-sections (Figure 7). We then interpolated P and T axes plunges in

the vertical 2D cross-section. The 1-sigma uncertainties of the P and T axes plunges probability

distributions are displayed in Figure S3 and S4 respectively. The cross-sections uncertainties

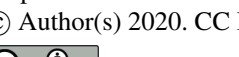



show that in the overall the localizations of the deformation patches are well resolved, while their exact spatial extent is less constrained.

Cross-section 1 from the Swiss molasse basin to the North of the Po plain (Figure 9a) comprises
153 events projected from 25 km on each side of the profile with depths in the 5-12 km range. Their inversion shows a majority of strike-slip mechanisms, with a pattern of extension localized in the inner part of the chain in the Southern Valais area, east of the CPF trace (cluster of events and extensive deformation mode already described in previous studies, see for instance Maurer et al. (1997); Eva et. al (1998)). A small extensional zone appears at 10 km
depth under the Northern Valais seismic zone, encompassed in a major transcurrent signal. These results are consistent with the inversion obtained at the surface. The extension appears to be relatively shallow on this cross-section, as it develops from 10 km depth up to the surface.

The second interpolated cross-section is located more or less along the ECORS profile (section 4.1; Marchant and Stampfli (1997)). It is constrained by 218 events distributed over 25 km on
each side of the profile. The focal depths are quite shallow (*i.e.* approximately 10 km) in the western part of the section and increases significantly eastward, at around 140 km along the profile, from 10 up to 30 km deep. This section shows two extensional areas, one in the shallow part corresponding to the Briançonnais arc east of the CPF, and the other one, around 20 km deep, corresponding to the northern tip of the Piémontais arc beneath the DM massif, separated
by an area of strike-slip regime, as shown in the map inversion (Figure 7). This section reveals a striking juxtaposition of extension and compression at 190 km along the profile. This compression pattern lies underneath the Po plain at a depth of 20 to 25 km. The occurrence of this unique compressive pattern appears robust from Figures S3 and S4, even if the juxtaposition limit between the extensive Piémontais pattern and the compression area seems
less constrained.

The third cross-section runs from the Pelvoux massif ("P", external zone) to the Po plain. It is made of 302 events up to 20 km away from the profile line, which corresponds to the width of deformation patches seen on the map (Figure 7). It also shows a deepening of the seismicity from West to East, from 10 to 30 km. The vertical interpolation gives a quite different view of
the extensional pattern compared to the map interpolation. Indeed, the map view (Figure 7) shows two extensive patterns separated by a narrow strike-slip strip along the Briançonnais arc. The cross-section however reveals that this extension is rather continuous at depth, with a shallower strike-slip pattern on the top centered on the CPF.






Figure 9. Vertical cross-sections displaying averages of P, T and B distributions from the Bayesian interpolation of events projected along 5 profiles. Profile locations are indicated on Figure 7. Labels refer to geological features displayed on Figure 3.

These features appear robust from the standard deviation cross-sections (Figure S3c and S4c). This strike-slip pattern located in the Briançonnais area seems to correspond to a local transcurrent zone in a regional extensional area. In continuity with cross-section 2, the compressional zone is located directly next to the extensional area, with a sharp boundary at depth (10-25 km), while the respective shapes of the compressional and extensive patterns are poorly resolved (Figure S3 and S4).

Cross-section 4 runs from the Provence area to the South of the Po plain and regroups 245 events located up to 15 km away from the profile. This profile is designed tighter than the other ones due to the higher heterogeneity in the focal mechanisms at the surface (Figure 7). The vertical interpolation is consistent with the map interpolation and reveals that the extensive pattern seen on the Piémontais arc is rather deep (from 12 to 20 km deep) and is surrounded by predominant strike-slip regime all around, and more specifically at shallower levels. The extensive area located at depth under the Piémontais arc appears robust; however, its lateral extension, deep under the Argentera massif and the Po plain seems poorly constrained.

Cross-section 5 is drawn along the strike of the belt, and runs from the Leman lake (Chablais area) to the region of Nice. It gathers 1186 events spread over 25 km on each side of the profile. This section aims at investigating the connections between the different extensional patches observed in the map interpolation. It appears that extension is characterized by a slightly increasing depth from North to South, while strike-slip remains dominant in the whole upper crust of the northern external zone and in the shallow levels to the South. Indeed, in this specific section, extension appears as a 200km-long strip dipping southward, while cutting through identical geological structures north and south of the profile. The extent of the deformation patterns is rather well resolved on this cross-section. Only the thin lineaments observed in the extensive area below the CPF and in the surrounding strike-slip areas appear to be artefacts.

## 4 Discussion

### 4.1 Comparative analysis of surficial and deep seismic deformation

The reconstructed maps of deformation styles presented in section 3.3 permits to address the structural control on the seismicity distribution and its deformation mode. From seismicity



maps and focal mechanism distributions, extension first appears continuous along the Briançonnais and Piémontais seismic arcs. However, once interpolated, strike-slip to transtensive regimes are not only prevailing at the periphery of the chain but also at several places intersecting the Briançonnais seismic arc, such as in the Northern Briançonnais and

Argentera, in association to significant energy release. Thus extension appears rather concentrated in several places along the belt in the southern Valais, in the Briançonnais, and in the Piémontais areas. Cross-sections views of the focal mechanisms distribution on the contrary seem to indicate transverse clusters of seismic events located beneath the Briançonnais and Piémontais arcs and the Po plain (cross-sections 2, 3 and 4 of Figure 9), sometimes undergoing

a continuous extensive regime from west to east (cross-sections 3 and 4). These discrepancies are due to the vertical integration of all focal mechanisms when working on map views. In particular, it hides the vertical stratification of the tectonic regime, where strike-slip events often occur above the extensive ones. This is well exemplified by the Piémontais extensive events (cross-sections 2 and 4).

The link between these deformation patterns and structural information is investigated in Figure 10, where we show the location of our focal mechanisms at depth superposed on top of two geophysical profiles. One profile is based on seismic reflection (ECORS-CROP, Marchant and Stampfli (1997)), and the second on local earthquake tomography (Solarino et al., 2018). Our focal mechanisms appear to follow the main horizontal reflectors along the ECORS-CROP

profile (Figure 10a), with their depth increasing from west to east. The seismic events seem grouped in several clusters along these two transverse profiles, especially beneath the Belledonne area and the Briançonnais and Piémontais arcs. Both profiles highlight the limits between European and Adriatic upper crust, with a wedge of Adriatic mantle (the so-called Ivrea body, *e.g.* Lyon-Caen and Molnar (1989); Paul et al. (2001)) located at the boundary

between the two crustal entities. Our focal mechanisms are localized in the upper crust. While the depth distribution of the seismic events follow the structure of the European crust (Figure 10b), the distribution of the style of deformation does not appear controlled by the structure of the former slab, both extensive and strike-slip events being imaged on the same sub-horizontal structure. However a few events are found deeper in the mantle wedge (Malusà et al., 2017),

which correspond to the compressive pattern observed in the Po plain (Figure 10b). The limit between the Piémontais extensive and the Po compressive patterns thus coincides with the boundary between the former European slab and the Adriatic crust. This is illustrated by the geometry of the Moho (Spada et al., 2017) on top of our map of interpolated deformation modes





(Figure 10c), where the limit between the European and the Adriatic Moho separates the
Piémontais extensive zone from the Po compressive one (Figure 10c).





Figure 10. a) Cross-section of computed focal mechanisms superposed on an ECORS-CROP seismic reflection profile, modified from Marchant and Stampfli (1997). Labels refer to geological features displayed on Figure 3. b) Focal mechanism cross-section superposed on a
CIFALPS profile of local earthquake tomography, modified from Solarino et al. (2018). White contours are isolines of equal Vp/Vs, solid lines represent main tectonic features, and bold letters refer to different regions of the model discussed in Solarino et al. (2018). Dashed lines represent the European (orange area around letter k to j) and Adriatic crusts (orange areas around letters h to g) limits, indented by a serpentinized mantle wedge (yellow area between
letters j and h). The Adriatic Moho (thick dashed line below letters h to g) is deflected west of the Po plain (~15 km depth) at the location of the Ivrea body gravimetric anomaly. Other labels refer to the geological features displayed on Figure 3. c) Contours of Moho depth (in km) from Spada et al. (2013) superposed on the map of deformation style from Figure 7. The European Moho (to the west), the Adriatic Moho (to the east), and the Ligurian Moho (to the south) are
separated by thick black lines. The location of the seismic profiles a) and b) are indicated by dashed lines.

**4.2 Implications on Alpine geodynamics**

This paper is focused on the analysis of an unprecedented number of focal mechanisms around the bend of the Western Alps and its connection with the Central Alps. We used state of the art
methods plus an innovative statistical approach to provide an updated 3D high-resolution view of the seismic deformation in this mountain belt. The unprecedented resolution of our analysis sheds new light regarding several detailed geodynamical aspects.

The current Alpine stress field and its related deformation mode have been investigated using inversion methods of focal mechanisms since the end of the 1990s (e.g. Baroux et al., 2001;
Kastrup et al., 2004; Maurer et al., 1997) with an increasing accuracy. Actually, these studies dealt with more or less local/regional areas, and only Delacou et al. (2004) addressed the problem at the scale of the entire Western and Central Alps. All these previous surveys relied on a limited number of focal mechanisms with respect to the present investigation and they systematically were inverted within zones expected to be homogeneous in terms of deformation
style, based on structural criteria. One of the main improvements of the present paper is the stress inversion using a grid strategy (section 3.2), which allows to get rid of any tectonic *a priori*. Notwithstanding these limitations, former stress inversions in the Alps have established a first order contrasted stress field. It is characterized by roughly orogen-perpendicular extension all along the backbone of the arc, surrounded by a transcurrent stress state at the
periphery of the orogeny, locally modulated by a reverse component. These features remain everywhere compatible with a dextral movement along the longitudinal Alpine strike. Our new high-resolution imaging of the stress field around the Alpine arc injects important modulations into this rough scheme. Firstly, transcurrent tectonism appears robust and much more important



in the whole Alpine realm than previously thought, in association with significant energy
release. This point is very important in terms of geodynamic interpretations and could not be
properly imaged by the previous 2D analyses (Delacou et. al (2004)). Secondly, we confirm the
prevalence of extension in the core of the Alpine arc, but we point out a more complex scheme
for the extensional zones (Figure 7 and Figure 9) in 3D, which appear now as extensional
patches embedded within an overall transcurrent field (Figure 7). These patches are located all
along the so-called Briançonnais and Piémontais seismic arcs, but the continuity of the
extensional area is no more supported by the inversion, as we retrieved 4 main zones of
extension more or less disconnected one from the other (Figure 7). The in-depth geometry of
the extensional zones is also revealed for the first time by our inversion of the deformation
mode (Figure 9). Thirdly, the direction of the principal stress axes in the internal zones (namely
the extensional σ3 axes) is systematically deflected of 30° to 40° clockwise from the radial
extension pointed out up to now. Moreover, we retrieved fewer and smaller local compressive
areas as shown by the previous studies (e.g. Delacou et al. (2004); Eva et al. (1997)). The only
significant zone of shortening appears to be located in the Po plain. We also found a small zone
of compression at the western front (Belledonne zone), but it is poorly constrained. Although
some individual reverse focal mechanisms are noticed to the South of the arc (Mercantour and
Var regions, Figure 2 and Figure 3), neither the deformation mode inversion nor the stress
inversion pointed out a reliable stable compression in this zone, in contrast to the interpretation
of Delacou et al. (2004). This may arise from the lack of data in the Ligurian Sea and
surroundings in our dataset (for this specific zone see for instance Béthoux et al. (1992);
Larroque et al. (2016)). The compressive pattern we obtained in the Po plain is, however, stable
and sharply juxtaposed with extension at the limit between Adriatic and European plates.

Beyond the seismotectonic approach *sensu stricto* (*i.e.* focal mechanisms analyses), the
comparison of the seismotectonic-related deformation with the one mapped using geodesy can
still enhance our comprehension of the Alpine geodynamics. The horizontal geodetic
deformation in the Alps is well correlated with the seismotectonic one in terms of deformation
style and of orientations (Figure 11; Walpersdorf et al. (2018)). In particular, the extensional
tectonics observed in the core of the belt are now generally identified by recent GNSS studies
(Masson et al., 2019; Mathey et al., 2020; Sánchez et al., 2018; Walpersdorf et al., 2015).
However, vertical motions seen by GNSS yield a regional uplift roughly correlated to the
Alpine topography in the northern and central part of the Western Alps, possibly up to 1 to 2
mm/yr (e.g. Serpelloni et al., 2013; Nocquet et al., 2016; although the magnitude can be





disputed, see Husson et al., 2018). Our precise mapping of the seismic extension compared to GNSS uplift shows that the patterns of extension and uplift are spatially uncorrelated, especially in the southern branch of the Alpine arc (Figure 11). Only two patterns of extension of our
interpolated deformation map (in the Northern Briançonnais seismic arc BRN and in the Southern Valais VSS) collocate with uplift. The extension located along the Piémontais arc is on the contrary associated with a zone of subsidence. Therefore, we state that uplift and extension, which were sometimes considered correlated (Champagnac et al., 2007; Vernant et al., 2013), are partially disconnected.

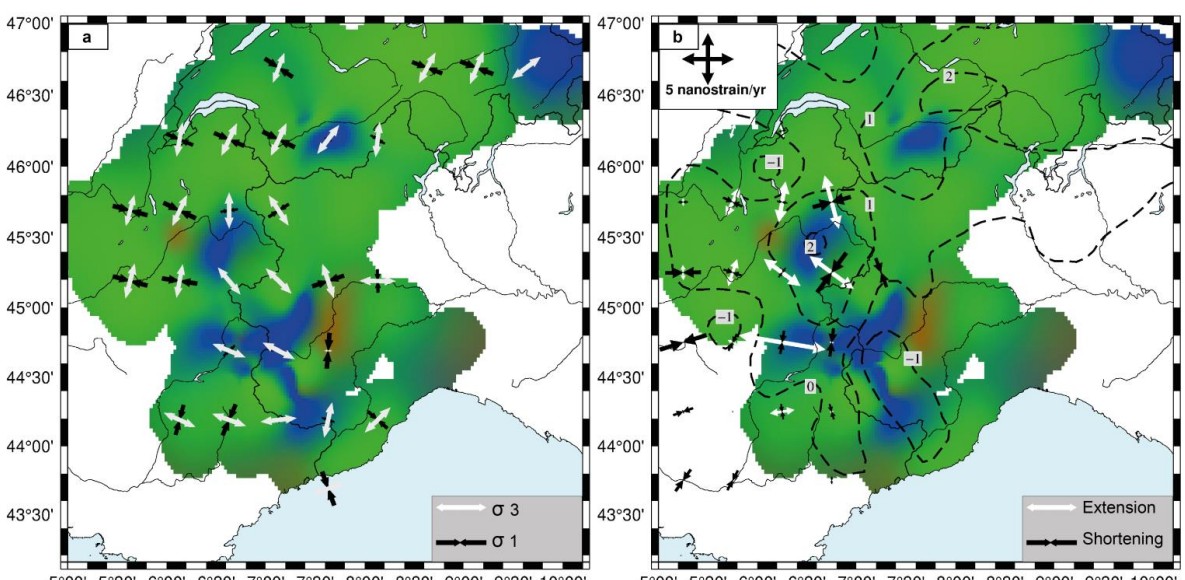

Figure 11. a) Stress field from MSATSI inversion from Figure 6. b) GNSS strain rates from Walpersdorf et al. (2018) showing extension (white arrows) and compression (black arrows). Dashed lines represent contours of vertical velocities (in mm/yr) from Sternai et al. (2019). In the background of both graphs is the Bayesian interpolation of deformation style from focal mechanisms from Figure 7.

In terms of geodynamic, the question of the processes driving the current deformation of the Alps remains a matter of debate. The prevalence of extension in the core of the belt, reinforced mounting evidence of geodetic uplift of the elevated regions of the Western Alps, led to the development of a series of models, alternatively involving intrinsic forces due to crustal and/or lithospheric roots, and extrinsic processes (see Sternai et al. (2019) for a review).
Extrinsic processes include the glacial isostatic adjustment (GIA; e.g. Gudmundsson (1994); Mey et al. (2016); Chéry et al. (2016)), erosion (e.g. Champagnac et al., 2007; Sternai et al.,



2012). According to Sternai et al. (2019) they could explain all together 50 to 70% of the uplift rates observed in the Alps. Besides isostatic adjustment to crustal deformation, intrinsic processes are related to deep dynamics in the Alpine lithosphere and predictions largely rely on the knowledge of the thermo-mechanical properties of deep structures. Slab dynamics in particular (see Lippitsch et al. (2003); Zhao et al. (2015); Kästle et al. (2018)) can cause transient dynamic topography (e.g., Faccenna and Becker, 2020), or influence exhumation processes (Baran et al., 2014; Fox et al., 2015).

All these mechanisms may, jointly or independently, explain the high frequency variations of the seismicity and kinematics. Yet, they can't account for the low noise transcurrent motion that prevails over the entire region. This strain field requires a more global geodynamic cause, and far field forces need to be invoked. Plate tectonics set the stage for local Alpine tectonics. The counterclockwise rotation of Adria with respect to stable Europe (e.g. Calais et al., 2002; Serpelloni et al., 2005, 2007) largely counterbalance buoyancy forces (Delacou et al., 2005), but may also add another component. While a purely plate-related geodynamic model seems discarded by now (D'Agostino et al., 2008; Devoti et al., 2008) due to the evidence of both extension and uplift in many places all along the Western Alpine arc, our observations may revive the role of plate motion in an attempt to explain the current Alpine kinematics and seismicity. Indeed, in addition to the predominance of transcurrent tectonics that we point out at the scale of the whole Western and Central Alps and forelands, we show a deflection of the extension axes in the core of the belt with respect to simple orogen-perpendicular extension. Both the transcurrent stress field at the periphery of the orogen, and the deflection of the extensional direction in the inner zones could be driven by counterclockwise rotation of Adria with respect to Europe. According to this scenario, buoyancy forces set the high frequency pattern of extension and compression, while far-field forces control the overall transcurrent field. Their joint effect interact to produce the complex deformation pattern and stress field revealed here.

**5 Conclusions**

The three-dimensional analysis of the mode and orientation of seismic deformation within the Western Alps provides, for the first time, a spatial resolution allowing to get rid of *a priori* seismotectonic zonation. The exceptional density of seismic data acquired within the Western Alps for the past 25 years allowed us to derive focal mechanisms for low magnitude events that were, to date, unavailable. From this new dataset we inferred a continuous seismic deformation



field characterized by the deformation style, at the surface and at depth, associated with an up-
to-date objective surface stress field. The resulting seismic deformation field is overall
consistent with the deformation patterns retrieved by previous seismotectonics studies as well
as with the horizontal geodetic observations. At depth, the distribution of computed focal
mechanisms matches the main reflectors already unveiled by seismic imagery. Our results also
highlight new features that could be observed only due to the increased spatial resolution we

provided. Most importantly, the probabilistic inversion of focal mechanisms points out a robust
predominant strike-slip regime at the regional scale, with dextral motion consistent with the
strike of the belt, and robust discontinuous extensive patches in the core of the belt, while
restricting robust compression to the Po plain and western Alpine front. The pattern of
deformation at depth, raises up the issue of the continuity of the extension pattern, as patches

of extension appear clustered along the two main seismic arcs. Extensional direction are in good
agreement with long term geodetic strain rates, and appear robustly oblique to the arc from
three inversion methods. The highly resolved seismotectonic regime sheds a new light on the
current dynamics of the alpine orogen, wherein far field plate tectonics, linked to the
counterclockwise rotation of Adria with respect to Europe imposes a global transcurrent regime,

while buoyancy forces explain the high frequency variations of extension and marginal
compression in the core of the belt.

**Data availability**

All the focal mechanisms computed for this study are available in .pdf format and in digital

format (.txt) in the supplementary materials.

**Acknowledgments**

This work was funded by both the IRSN (Institut de Radioprotection et de Sûreté Nucléaire)
and the LabEx OSUG@2020 (Investissement d'avenir – ANR10LABX56). The authors would

like to thank the Sismalp team for maintaining the regional seismic observation network. This
work contributes to an effort of valorization of long-term seismic and geodetic observations in
the framework of the RESIF-EPOS National Research Infrastructure (Réseau Sismologique et
géodésique Français, doi:10.15778/RESIF.FR).

**Supplementary Materials**

*Supplementary_SE_Mathey.pdf* contains figures S1, S2, S3 and S4 and table S1



*liste_meca.txt* contains table S1 in digital format

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
