# Peer review of "Present-day geodynamics of the Western Alps: new insights from earthquake mechanisms"

_Solid Earth, 2020_

## Referee Comment (RC1) · Anonymous Referee #1 · 29 Dec 2020

Dear editor, I have read the paper Present-day geodynamics of the Western Alps: new insights from earthquake mechanisms by Marguerite Mathey et al. and I am now ready to share my suggestions. You can pass this comments to the authors. The paper is based on a large dataset of focal mechanisms in the Western Alps that are used for the computation of seismic stress and strain, later apparently intepreted in a geodynamic perspective. Despite the number of data used, I have several concerns on the quality of the dataset and, as a consequence, on the reliability of the obtained results. Moreover, I don't see any geodynamic interpretation, but a seismotectonic one. I also believe that the references are not up to date and not adequately taken into account in the discussion. Therefore, in my opinion, the paper is not ready for

publication in the present form and needs major revision. I am available for a second round of review. Coming to detail, the authors use 2215 focal mechanisms that, in my opinion, are selected with not enough restrictive criteria. In particular. the maximum allowed azimuthal gap of 90° between polarities and the maximum azimuthal gap of 60° for incidence angles is not sufficient to constrain the quality of a focal mechanism. With such a threshold the dataset is, according to the description reported by the authors, made of only 0.7% of data with quality A; 2.3% have a quality B, 3.2c% have a quality C while the majority of focal mechanisms (93.8%) have a quality D. These latter events have an uncertainty on the strike of 45° to 55°, which is a lot given that this uncertainty also likely affects the other focal parameters. However, it looks like all data are treated as equal quality in the computations. In my opinion, the authors should improve the quality of the dataset or discharge the lower quality events. One more issue is that, according to what the authors state, they make an arbitrary choice of focal solution ("A preferred focal mechanism was computed for each of the 2,215 events meeting these criteria"). It is necessary to describe on which criteria the selection is made. A constrained focal solution does not require choices. I have also issues about how the authors could recognize clear polarities on waveforms for events with magnitude lower than 1.0 (in my experience, it is even hard to do it on 2.0 and above magnitude events). Anyone dealing with seismogram phase pickings knwo that the S/N ratio of most seismic stations does not allow to push down to smaller magnitudes. However, some examples, added to the supplementary material, could help the reader evaluate the quality of the polarity readings and thus of the focal solutions.

Minor issues Line 90 "by six local or national networks operated from 1989 to 2014" Only five are listed

Line 105 "Arrival time uncertainties were harmonized." How ? "Potential picking errors were identified and cleared out." How ?

Line 110 "The complete set of earthquakes includes blasts, quarrying or mining events." I do not see the significance of plotting non-natural seismicity. Please discharge these

events from the dataset

Line 120 "Thanks to the high density of stations provided by the combination of six networks, we were able to apply strict computation criteria. The maximum allowed azimuthal gap between polarities was set to 90° and the maximum azimuthal gap of incidence angles to 60°." As already discussed, these criteria are not strict at all.

Line 175 "Strain rates are computed by averaging moment tensors (i.e., symmetrical 9 components 2nd order tensor, plus seismic moment amplitude), for which the 9 components directly depend on strike, dip and rake parameters of the focal mechanisms." That means that the quality of focal mechanisms is fundamental. . ...

Line 315 "we investigated the distribution of stress orientations using focal mechanism inversions. All inverted earthquakes are equally weighted, regardless of their magnitude" If all focal mechanisms are equally weighted their quality is not considered

Line 375 "The angle between the direction of extension and the strike of the belt is of the order of 30° to 40°." I disagree, northern of GP is almost perpendicular.

Line 380 "All the western periphery of the belt (corresponding to the zones VSN, DPH, NMT,VAR Figure 5) show strike-slip stress fields, with a rotating state of stress compatible with dextral motions along longitudinal directions (typically along longitudinal faults such as the Belledonne fault, Thouvenot et al., 2003). " Ok for VSN and DPH why for the other sectors?

Fig. 9 Named cross-section in the text as in the fgure 9

Line 535 "The seismic events seem grouped in several clusters along these two transverse profiles" Where are seismic events in figure 10a? Do you mean the focal mechanisms? The focal mechanisms cannot have clusters, it depends on which earthquakes you choose to compute focal mechanisms.

Line 540 "While the depth distribution of the seismic events follow the structure of the European crust (Figure 10b)," Where are seismic events in figure 10b?

Line 585 "former stress inversions in the Alps have established a first order contrasted stress field. It is characterized by roughly orogen-perpendicular extension all along the backbone of the arc, surrounded by a transcurrent stress state at the periphery of the orogeny, locally modulated by a reverse component." Which authors?

Line 600 "Thirdly, the direction of the principal stress axes in the internal zones (namely 600 the extensional $\sigma$3 axes) is systematically deflected of 30° to 40° clockwise from the radial extension pointed out up to now." I disagree.

---

## Referee Comment (RC2) · Anonymous Referee #2 · 19 Jan 2021

Summary and General Comments:

To improve the understanding of present-day seismotectonic processes in the Western Alpine arc, Mathey et al. computed a new catalogue of first-motion focal mechanisms (FM). This FM catalogue is then used to analyse seismic strain and corresponding orientations and regimes of tectonic stress using various inversion and regression methods. The new set of focal mechanisms is derived from an existing compilation of relocated hypocenters in the Western Alps, with corresponding take-off angles and bulletin first-motion polarities of P-waves. The methods for computing focal mechanisms as well as their analysis (including the computation of strain-rates and stress-inversions)

[Figure]

are largely based on established, commonly used methods and software tools. According to the authors, the number of FMs derived in this study is larger than any existing FM catalogue for the region and thus allowing continuous imaging of deformation regimes in 3-D at unprecedented resolution. Their main findings are: i) although consistent to first-order with previous results, fine-scale variations in the extensional regime ("discontinuous extensive patches") in the core of the Western Alps are imaged. ii) predominantly strike-slip regime in most other parts of the arc. iii) N-S compressional regime at the eastern boundary towards the Po plain and (less pronounced) along the western Alpine front. iv) the extensional direction is oriented oblique with respect to the strike of the Western Alpine arc (instead of perpendicular). v) Pattern is likely explained by a superposition of far-field horizontal stresses due to rotation of Adria and vertical deformation due to buoyancy forces.

The manuscript is well organised and written and analysing methods and presentation of results appear reasonable. However, as summarized in the next section, my main concern relates to the quality-control of the derived FM catalogue and possible impacts on the derived conclusions. In addition, the discussion of the imaged 3-D deformation pattern is general and vague in some places and should be improved and extended as outlined below. In summary, I recommend major revision of the manuscript.

Specific comments:

1) Quality of FM-Quality: I have serious concerns related to the quality-control of the presented FM catalogue. About 1258 solutions (out of the 2215 total FMs) have ML magnitudes <2.0. In my experience, working on FMs in the Alpine region, unique, high-quality FM solutions in this magnitude range constrained from P-polarities alone are extremely rare. In almost all cases the solutions are highly uncertain and ambiguous and -if at all- only possible in regions of extremely dense station coverage. In addition, polarities in standard, routinely picked bulletin data (as used in this study) are full of mistakes/blunders and not very reliable unless reviewed by experienced seismologists with the goal to derive a reliable focal mechanism. This is likely even worse

when data is combined from different bulletins. In combination, I expect severe uncertainties and ambiguities in the derived FMs. In less dense instrumented areas, I would even consider solutions in the ML range 2.0-3.0 (769 FMs) as highly uncertain (at least partly), e.g. if the focal depth is poorly constrained and therefore take-off angles can be highly uncertain. In the provided FM catalogue (supplementary material) 94% of the FM solutions are of lowest quality class "D" (2076 FMs) indicating that the majority of the solutions is highly uncertain. The authors associate "D" qualities with "strike uncertainties" of 45-55 deg (and I assume it's even higher for dip and rake). It seems, however, that the a priori quality of the solutions is not taken into account in any of the applied inversion schemes. The uncertainties derived by the Bayesian inversion of P and T plunges in Fig. 8, on the other hand, are surprisingly small (<10 deg for the majority of the region, values I would associate with high-quality FM solutions). This uncertainty in P and T plunge seems therefore unrealistically small to me given the potential quality of the majority of the used "D" FM solutions and raises the question how much of the "small-scale variations" in the deformation regime is related to noise introduced by the large amount of low-quality FM solutions. The authors should extend their discussion on the potential uncertainties of the low-quality solutions and how it potentially affects their inversion results. E.g. they could use a priori weighting based on the solution qualities in their inversion and compare results against solutions without quality weighting. Also, in some of the maps, the locations of higher-quality solutions could be marked by circles of different colour to identify the regions which are constrained by more reliable solutions. In addition, I also miss a benchmark of the "automatically" FM solutions derived by the authors against existing, manually reviewed, high-quality FM or moment-tensor solutions published by French, Italian and Swiss agencies. This would help to assess the potential uncertainties and reliability of this catalogue.

2) Seismic strain rates: As the authors stated themselves, the seismic moment is dominated by the largest events in a region. I am therefore wondering how much the seismic strain rates in Figure 4 and the seismic "flux" in Fig. 7a are controlled by the largest events in this region. I would therefore suggest to plot the corresponding beachballs

of events with ML>=4.0 in the background of Figure 4 and indicate the corresponding magnitude of these events. This might help the reader to understand if the corresponding strain regime and strain rate in the corresponding zonation is mainly dominated by the largest event within this zone. Is the relatively smaller moment release in the NE corner of Figure 7a real or just an artefact of the completeness of the author's catalogue? In Figure 2c, I see several M3-4 events in this region and I would have assumed to see a corresponding relative increase in moment release in this part as well.

3) Discussion/Interpretation: In my opinion, the discussion part could be extended and improved. The discussion is a bit vague and general. Some aspects which could be extended:

- Role of slab dynamics in the Western Alps: Different models have been proposed on the slab structure: detached (e.g. Lippitsch et al 2003) vs. attached (e.g. Zhao et al 2016). In addition, if attached, rollback and delamination might play a role (e.g. Sue et al 1999). How do the latest insights into slab-models fit/support your observations? Do the presented results add new constraints to this ongoing discussion?

- I do not understand how robust (and how to interpret) the N-S compression in the western Po plain shown in Figure 6 is. In Figure 5 this compression seems more SW-NE? and others (Delacou et al 2004, Eva et al 2020 (tectonics)) show also more SW-NE or even E-W directed compression. Is this due to very few mechanisms of one cluster? How well are they constrained? Why did the former compressional domains east and west of the Arc (e.g. Delacou et al 2004) "disappear/reduce" in your results? Is it possible that the weight of the compressional events is just suppressed by the larger number of additional small (and likely noisy) low-quality FM solutions of oblique type? Or did the former reverse type mechanisms change to strike-slip in your FM calculation? If so, are the new solutions better constrained then the previous solutions?

- How can the contrasting juxtaposition of compressive and extensional domains at the eastern boundary be explained (e.g. Figure 9c)? Similarly, can the mix of strike

slip and normal mechanisms within the core of the Western Alps be simply explained by a general transtensive regime (some faults take up strike-slip, others the normal component)? Are principal axis compatible with transtension (i.e. is the orientation of the T-axis consistent and only B and P flip?). If not, it is difficult to explain with a homogeneous far-field stress. A final sketch figure indicating the overall kinematics would certainly help to summarize and better explain the proposed model of vertical deformation controlling the patches of extension while rotating Adria controls the overall strike-slip regime. Still the question remains why there is little spatial correlation between "extensional patches" and surface uplift. Also, you should compare your results and model in more detail with the recent paper of Eva et al 2020 (Tectonics), who propose a similar 3D seismotectonic model of the same region.

4) The English is reasonable; however, I spotted some smaller issues I have listed in the following list. I would recommend another round of proofreading by the authors.

Detailed Comments:

- Line 33: "lays" -> "locates"

- Line 34 and many other places in the manuscript: "high frequency spatial variations" -> I would replace "high frequency" with e.g. "short wavelength" or "variations over short distances". Frequency relates to time not space.

- Line 43: "alpine belt" -> "Alpine belt"

- Line 48: "with respect to Europe" or "the European plate"

- L. 52: "Western Alpine arc"

- L. 71: "updated catalogues allow to"

- L. 80: Indicate the exact period of the catalogue: from 19XX to 20XX. What is the estimated Mc of this catalogue? All events included from all contributing agencies or only above a certain magnitude?

- Figure 1: Maybe here (or in another map) highlight the location of M>=4 events, e.g. as circles of different colour.

- L. 100: SED operates currently >200 stations. For a more up-to-date reference on the network see e.g. Diehl et al. 2018, Earthquakes in Switzerland and surrounding regions during 2015 and 2016, https://doi.org/10.1007/s00015-017-0295-y

- L. 102: "recorded" -> "installed"

- L. 108: "combined a 3D-velocity inversion"

- L.110: "in the order of a few degrees" – This is quite an optimistic statement in my experience. Especially for shallow sources, the uncertainty in focal depth can result in quite significant uncertainties in take-off angles, which can be more than few degrees... Not sure how this uncertainty is estimated, but I would at least add "for well constrained hypocenters"...

- L. 111: An earthquake is not a blast, I would write: "The complete catalogue includes earthquakes as well as blasts related to quarry and mining activities."

- L. 119: What is the S-phase needed for? Are you using S-polarities as well? As pointed out above, the azimuthal/take-off gap criteria are not sufficient to exclude grossly ambiguous FM solutions. I would have rather considered HASH's number of FM families (use only solutions with 1 family of solutions -> +/- unique solution) and number of possible solutions as quality criteria.

- L. 126: "uncertainties on picking" -> shouldn't it be "in polarities"? Did you use quality weights for polarities (e.g. U/D vs +/-) for the grid search? Did you use the uncertainty in TOA? How is this uncertainty estimated and how is it implemented in your HASH procedure?

- L. 127: Was ML recomputed by Potin or is this basically the original SISMALP ML? Is ML really always measured from S or is it simply the peak amplitude on the horizontal components (regardless of P,PmP, S, SmS, surface wave)?

- L. 143: "pression" -> "pressure"

- L. 144: "The representation of style . . ." Rephrase this sentence.

- L. 180: This is a bit confusing, first the authors summarize different scaling-relationships to convert Ml to Mw, referring to a study proposing a polynomial fit but then it seems the authors use a 1:1 relationship without too much explanation why this is valid. What is the mistake in term of Mo of this simplification for larger events (M>4.0)? Why not using the already established relationship?

-L. 200: In classic stress-inversion the slip is assumed to be on the active plane (not on the auxiliary one). Others (e.g. Kastrup et al 2004) therefore down-weighted solutions for which the active plane is unknown (e.g. no information from relative relocation available). How did the authors address this problem in their stress-inversion strategy?

-L. 243: The methods "enables us to assess whether formal uncertainties on fault planes . . . are over or underdetermined". What is the result of this assessment then for your data? What are the formal errors of your focal mechanisms?

- L. 267: "in each" -> "for each"?

- L. 286: "are presented as surface projections in Fig. . ."

- L. 289: -> "Therefore, including historical events . . . in the summation could modify. . ."

- Figure 5: I see some differences in P and T between the two inversion methods which seem larger than the 1sigma estimated by the Bayesian inversion. Any explanation for these differences?

- Line 344: You mean both methods give similar results for the orientation of sigma3? Consider rephrasing this sentence.

- L. 351: -> "zoning realised" -> "zonation defined . . . contains seismicity of different tectonic regimes"

- L. 354: -> "Overall, the set. . ."

- L. 381: Not sure I understand this sentence. What are "longitudinal directions/faults"? Isn't the mentioned fault striking NE-SW?

- Figure 7b) why not add a colour bar (similar to Delacou et al 2004) rather than describing the meaning of the colours in the caption?

- L. 455 and elsewhere in this section: Instead of "153 events" I would write "153 focal mechanisms".

- L. 459: This extensional zone at 10 km: How well is the depth constrained for the associated events? This zone north of the Valais is more complicated than strike-slip. It contains all kinds of mechanisms: Strike-slip, oblique normal, oblique reverse, most likely it consists of an array of strike slip faults connected by releasing and restraining bends/step-overs. See e.g. Diehl et al. 2018, Earthquakes in Switzerland and surrounding regions during 2015 and 2016, https://doi.org/10.1007/s00015-017-0295-y. A bit north of the major strike-slip zone, towards the Alpine front, there is indeed a zone of extensional events (e.g. M4.3 Chateau-D'Oex earthquake of 2017, Jaun M3.8 event of 1999), which is maybe what the authors image in their cross-section 1 (needs to be checked). It is described and discussed in a recent publication which is currently in press and should be published online soon (Diehl et al. 2018, Earthquakes in Switzerland and surrounding regions during 2017 and 2018, https://doi.org/10.1186/s00015-020-00382-2). However, this extensional domain is much shallower than in the author's cross-section (uppermost crystalline basement or Mesozoic sediments, likely <5 km).

- L. 470: -> "as shown in Figure 7b"

- L. 480: -> ". . . . Figure 7b shows . . ."

- Figure 9: It would be helpful to add additional geological reference information from geological profiles, like position Alpine front, Ivrea body, etc. What are the tiny black dots (difficult to see)? Projected earthquakes (everything) or just earthquakes with

corresponding focal mechanisms used in the inversion/regression? I would make the symbols corresponding to FMs bigger (maybe as circles), try colour code the quality of the mechanisms. This would help to distinguish parts well constrained by data from areas with inter- or extrapolated values.

- L. 502: -> "... runs from Lake Geneva ... "

- L. 507: remove "Indeed"

- Figure 7b/text around line 517: Why not show Figure 7b for different depth intervals (similar to tomographic results) rather than projecting everything to one layer? This would lead to a "patchier" distribution with more white-spaces, but would avoid some of the misunderstanding due to vertical projection?

- L. 540: "...follow the structure of the European crust..." Not sure in terms of what? You mean in terms of dip? Or lithology? Should be more specific.

- L. 543: "former slab" What do the authors mean here? Does this "former slab" relate to the possibly detached slab? As mentioned above, this discussion needs to be extended.

- Figure 10: Are the beachballs shown on the profile in b) lower hemisphere projections (as in map view) or cut along the profile (projections)? Since they all plot on each other it's difficult to see anything... In caption of 10b, why not simply say: Dashed lines represent the European and Adriatic Moho after (???). Moho in 10b is from Spada as well?

- L 585: "... which is independent of any a priori tectonic zonation"

- L 586: "... movement along the longitudinal Alpine strike" Not sure what the authors mean here.

- L. 588: "rough scheme" -> "first-order distribution"?

- L. 650: "low noise transcurrent motion" Low-noise in terms of what?

- L. 655: "but may also add another component" -> What is this other component?

- L. 655: "While a purely plate-related geodynamic model seems discarded by now. . . our observations may revive the role of plate motion. . ." This sentence doesn't make much sense to me. Is it discarded or not? What other process should explain seismicity and deformation?

- L. 678: "seismic imagery" -> "seismic imaging methods"

- L. 687: "The high spatial resolution of seismotectonic regimes sheds light. . ."

- L 695: Why not add used polarities and take-off/azimuth angles to the supplementary material to allow others to assess the quality of mechanisms.

- L 698: What about all the data added from other networks? Nowadays most networks have DOIs and should be cited with their corresponding network code and DOIs.
* * *

---

## Author Comment (AC1) · 22 Mar 2021

Reviewer 1:

General answers:

1) Quality of the dataset:

We thank both reviewers for their help in improving our manuscript. We answered thoroughly to all their suggestions and interrogations as detailed below, especially concerning the long lasting debate related to the use and representativity of small magnitude earthquakes and corresponding fault plane solutions.

To answer the main concern of reviewer 1 we would like to emphasize on a few aspects of our work. First, several studies of the past decades, (e.g. Maurer, 1993; Bonjer, 1997; Thouvenot et al., 2003, Diehl et al 2021) already demonstrated that small magnitude focal mechanisms could be well constrained and reliable, even for Ml < 1, if based on local seismic network records, which is the case of the Western-Alps-dedicated Sismalp network, which records are additionally completed by national networks. Second, several studies also demonstrated that these small magnitude earthquakes are as much representative of the regional tectonic regime as higher magnitude ones (e.g. Amelung and King, 1997, Kastrup et al., 2004 for the Alps), which is once again demonstrated here (Figure R1 and supplementary material).

Concerning the polarity readings, we stress that intensity is more important than magnitude in the matter of identifying clear deviation directions in the waveforms. In our case we deal with shallow earthquakes recorded by a dense local network, resulting in distances most of the time ~ 10 km for the closest station, as well as in short ray lengths, providing a good S/N ratio (e.g. Thouvenot and Bouchon, 2008, Thouvenot et al. 2003). Several studies focused on the Western Alps moreover showed that for specifically chosen low noise sites with stations less than 50 km away from hypocenters, reliable focal mechanisms could be derived for low magnitude events (e.g., aftershock sequence of the Vuache earthquake, Thouvenot et al., 1998).

However, we recognize that the unprecedented focal mechanisms dataset computed in the present study is likely noisy and bearing several mistakes both on polarity readings and on derived focal mechanisms. Despite the presence of less-well constrained mechanisms, we choose to use the whole dataset presented here since our study aims at increasing the spatial resolution of seismic stress and strain fields which were previously derived on the Western Alps from moderate magnitude earthquakes (e.g. Delacou et al., 2004, Kastrup et al., 2004, Sue et al., 2007, Bauve et al., 2014). Since very few moderate earthquakes occurred since then, this goal could only be achieved by using lower magnitude events. This initiative results from the increasing number of available seismic records for low deforming areas in the recent years, provided by seismic networks in constant evolution. For our study, we benefit from an unprecedented low magnitude seismic records dataset which we aim to exploit for seismotectonic purposes for the first time. To achieve our goal, we aim at exploring the statistical distribution of this database, since it does not present any systematic bias in polarity readings nor any correlated uncertainty in focal mechanisms. Thus, in addition to classical seismotectonic inversion procedures, we assess which features of the seismic deformation field are robust through a Bayesian inversion scheme in which focal mechanisms are weighted according to their quality.

Last, we stress that the selected focal solutions dataset was already cleared out from possible worse solutions by applying the computation criteria described in the main text section 2.2, resulting in A-D quality events instead of A-F quality solutions. We would also like to emphasize that the "D" quality flag outputted by HASH is rather conservative and represents formal uncertainties depending on several criteria such as the number of polarities and station distribution ratio, but does not necessarily represent an unreliable mechanism. We stress that we used in the main text the word "preferred mechanism" as defined by Hardebeck and Shearer (2002) : "The preferred solution, or the most probable solution, is the average of the acceptable fault plane solutions after outliers have been removed [...] If there are clustered outliers, alternative solutions (or "multiples") are found based on those outliers.". Thus it does not represent an arbitrary choice but a statistical one on which uncertainty estimates and quality flags are based. However, since we are aware that our dataset presents various quality events, we provide the full list of focal mechanisms computed along with their quality flag, so that the reader may choose to rely only on a subset of focal mechanisms to further investigate fault plane solutions for any specific area. In addition to this quality criteria, to additionally help the reader to decipher which events provide more constraints on each inversion result, we now provide in the revised version of the manuscript the magnitudes of earthquakes with Ml >3.5 included in our inversion procedures.

2) Geodynamics:
       In our view, the discussion of processes driving crustal deformation and seismicity represents a geodynamical discussion. We improved this discussion with a final interpretative sketch and by further discussing several geodynamical aspects detailed in the answer to reviewer 2, and added the corresponding citations to the reference list.

Minor issues
   ● *Line 90 "by six local or national networks operated from 1989 to 2014"*
*Only five are listed*
       This is now corrected in the main text.

   ● *Line 105 "Arrival time uncertainties were harmonized." How ? "Potential picking errors were identified and cleared out." How ?*
       Picking weights from 0 to 4 were translated into time uncertainties through a statistical analysis, and outliers were removed through Wadati diagrams analysis. We refer to Potin (2016) for further information on the data compilation procedure.

   ● *Line 110 "The complete set of earthquakes includes blasts, quarrying or mining events."*
*I do not see the significance of plotting non-natural seismicity. Please discharge these events from the dataset*
       Polarities are indeed not read for quarries. We discharged these events from the dataset (Figure 1 and Figure 7) in the revised version as requested.

   ● *Line 120 "Thanks to the high density of stations provided by the combination of six networks, we were able to apply strict computation criteria. The maximum allowed*

*azimuthal gap between polarities was set to 90 ◦ and the maximum azimuthal gap of*
*incidence angles to 60 ◦ .”* *As already discussed, these criteria are not strict at all.*

These criteria appear more strict than many displayed in the seismotectonic literature : e.g. minimum number of polarities of 6 and max azimuthal gap of 220° (Ammirati et al., 2019), maximum azimuthal gap = 180° (Sue et al., 1999) … We thus provide in our view strict criteria in order to take into account the specifically low deforming context of our study area. However we stress that, as mentioned above, the goal of our study is to provide an increased spatial resolution of the seismic deformation pattern, which can only be obtained by using a new, thus more noisy, focal mechanism dataset. For this exact reason we provide the complete list of the derived focal solutions with their associated quality flag in the case further studies would wish to focus on a specific area while using only higher quality focal mechanisms.

- *Line 175 “Strain rates are computed by averaging moment tensors (i.e., symmetrical 9 components 2nd order tensor, plus seismic moment amplitude), for which the 9 components directly depend on strike, dip and rake parameters of the focal mechanisms.”*
*That means that the quality of focal mechanisms is fundamental.*

As explained in the main text we are aware that the direction of the strain axes could be influenced by a few stronger earthquakes only. That is why we also compare the direction derived from strain rate estimates with the directions obtained through stress inversions.

- *Line 315 “we investigated the distribution of stress orientations using focal mechanism inversions. All inverted earthquakes are equally weighted, regardless of their magnitude”* *If all focal mechanisms are equally weighted their quality is not considered.*

As described section 2.2 (l. 137-141), we use several classical seismotectonic tools, in which focal mechanisms are either not weighted (stress inversions), weighted depending on their magnitude (Kostrov summation), or weighted according to their quality (Bayesian surface reconstruction). We thus compare the different results to assess the quality of our results. Our results suggest that small magnitude events carry as much regional tectonic signal as higher magnitude ones since results are, to the first order, similar between the different approaches.

- *Line 375 “The angle between the direction of extension and the strike of the belt is of the order of 30 ◦ to 40 ◦ .”* *I disagree, northern of GP is almost perpendicular.*

This result description was indeed only right to the first order. We replaced the corresponding sentence by the following more nuanced one in the revised version : "**The angle between the direction of extension and the strike of the belt is of the order of 30° to 40°, except north of Gran Paradiso massif (GP) where the sigma3- axis of the transtensional stress tensor is almost perpendicular to the strike of the belt. This shows that the extensional direction is almost systematically deflected clockwise with respect to the direction normal to the arc.**"

- *Line 380 “All the western periphery of the belt (corresponding to the zones VSN, DPH, NMT,VAR Figure 5) show strike-slip stress fields, with a rotating state of stress compatible with dextral motions along longitudinal directions (typically along longitudinal faults such as the Belledonne fault, Thouvenot et al., 2003). “* *Ok for VSN and DPH why for*

*the other sectors?*

In the Southern part, the western periphery of the chain includes several longitudinal right-lateral faults, such as the Sérenne, Tinée and Bersézio faults (Bauve et al., 2014), as well as transversal and antithetic NNE-SSW left-lateral faults (e.g. Blausasc fault, Courboulex et al. 2007) .

- *Fig. 9 Named cross-section in the text as in the fgure 9*
  References to Figures 9a) to 9e) were added in the main text as suggested.

- *Line 535 "The seismic events seem grouped in several clusters along these two transverse profiles" Where are seismic events in figure 10a? Do you mean the focal mechanisms? The focal mechanisms cannot have clusters, it depends on which earthquakes you choose to compute focal mechanisms.*
  This shortcut was corrected with the following sentence : "**Seismicity appears grouped in several patches of distinct deformation style along these two transverse profiles, especially beneath the Belledonne area and the Briançonnais and Piémontais arcs where the two main extensive patterns described in section 3.3 are lying.**"

- *Line 540 "While the depth distribution of the seismic events follow the structure of the European crust (Figure 10b)," Where are seismic events in figure 10b?*
  Same as above, the term "seismic events" was replaced by "focal mechanisms".

- *Line 585 "former stress inversions in the Alps have established a first order contrasted stress field. It is characterized by roughly orogen-perpendicular extension all along the backbone of the arc, surrounded by a transcurrent stress state at the periphery of the orogeny, locally modulated by a reverse component." Which authors?*
  We added the appropriate references : Maurer et al., 1997; Baroux et al., 2001, Sue et al., 2002; Delacou et al., 2004; Kastrup et al., 2004, Bauve et al., 2014.

- *Line 600 "Thirdly, the direction of the principal stress axes in the internal zones (namely 600 the extensional σ3 axes) is systematically deflected of 30◦ to 40◦ clockwise from the radial extension pointed out up to now." I disagree.*
  We nuanced this point of view as done above and rephrased the sentence : "**the direction of the least compressive stress axes in the internal zones (namely the σ3 axes) is almost systematically deflected of 30 ◦ to 40 ◦ clockwise from the orogen-perpendicular extension pointed out up to now.**"

[Figure]

Figure R1. Kaverina diagrams displaying the style of deformation of the computed focal mechanisms for different magnitude ranges.

**References:**

Amelung, F., & King, G. (1997). Large-scale tectonic deformation inferred from small earthquakes. *Nature*, *386*(6626), 702-705.

Ammirati, J. B., Vargas, G., Rebolledo, S., Abrahami, R., Potin, B., Leyton, F., & Ruiz, S. (2019). The crustal seismicity of the western Andean thrust (central Chile, 33°–34° S): Implications for regional tectonics and seismic hazard in the Santiago area. *Bulletin of the Seismological Society of America*, *109*(5), 1985-1999.

Bonjer, K. P. (1997). Seismicity pattern and style of seismic faulting at the eastern border fault of the southern Rhine Graben. *Tectonophysics*, *275*(1-3), 41-69.

Courboulex, F., Larroque, C., Deschamps, A., Kohrs-Sansorny, C., Gélis, C., Got, J. L., ... & Mondielli, P. (2007). Seismic hazard on the French Riviera: observations, interpretations and simulations. *Geophysical Journal International*, *170*(1), 387-400.

Diehl, T., Husen, S., Kissling, E., & Deichmann, N. (2009). High-resolution 3-DP-wave model of the Alpine crust. *Geophysical Journal International*, *179*(2), 1133-1147.

Diehl, T., Clinton, J., Deichmann, N., Cauzzi, C., Kästli, P., Kraft, T., ... & Wiemer, S. (2018). Earthquakes in Switzerland and surrounding regions during 2015 and 2016. *Swiss Journal of Geosciences*, *111*(1), 221-244.

Diehl, T., Clinton, J., Cauzzi, C., Kraft, T., Kästli, P., Deichmann, N., ... & Wiemer, S. (2021). Earthquakes in Switzerland and surrounding regions during 2017 and 2018. *Swiss Journal of Geosciences*, *114*(1), 1-29.

Kissling, E., Schmid, S. M., Lippitsch, R., Ansorge, J., & Fügenschuh, B. (2006). Lithosphere structure and tectonic evolution of the Alpine arc: new evidence from high-resolution teleseismic tomography. *Geological Society, London, Memoirs*, *32*(1), 129-145.

Maurer, H. (1993). *Seismotectonics and upper crustal structure in the western Swiss Alps* (Doctoral dissertation, ETH Zurich).

Piromallo, C., & Faccenna, C. (2004). How deep can we find the traces of Alpine subduction?. *Geophysical Research Letters*, *31*(6).

Salimbeni, S., Malusà, M. G., Zhao, L., Guillot, S., Pondrelli, S., Margheriti, L., ... & Zhu, R. (2018). Active and fossil mantle flows in the western Alpine region unravelled by seismic anisotropy analysis and high-resolution P wave tomography. *Tectonophysics*, *731*, 35-47.

Sue, C., Thouvenot, F., Fréchet, J., & Tricart, P. (1999). Widespread extension in the core of the western Alps revealed by earthquake analysis. *Journal of Geophysical Research: Solid Earth*, *104*(B11), 25611-25622.

Thouvenot, F., Fréchet, J., Tapponnier, P., Thomas, J. C., Le Brun, B., Ménard, G., ... & Hatzfeld, D. (1998). The ML 5.3 Epagny (French Alps) earthquake of 1996 July 15: a long-awaited event on the Vuache Fault. *Geophysical Journal International*, *135*(3), 876-892.

Thouvenot, F., & Bouchon, M. (2008). What is the Lowest Magnitude Threshold at Which an Earthquake can be Felt or Heard, or Objects Thrown into the Air?. In *Historical Seismology* (pp. 313-326). Springer, Dordrecht.

Zhao, L., Paul, A., Malusà, M. G., Xu, X., Zheng, T., Solarino, S., ... & Zhu, R. (2016). Continuity of the Alpine slab unraveled by high-resolution P wave tomography. *Journal of Geophysical Research: Solid Earth*, *121*(12), 8720-8737.

---

## Author Comment (AC2) · 22 Mar 2021

1) Quality:

We thank the anonymous reviewer for his constructive comments. As stated in the answer made to reviewer 1, reliable focal mechanisms could be computed for small magnitude earthquakes thanks to the dense regional Sismalp network and thanks to the good constraints provided by the relocation of the events in a 3D model and by the corresponding take-off angles. We acknowledge that, while we could not check visually each of the 2215 focal solutions selected for this study, not all of them are equally well constrained. However we would like to stress once more that the aim of our study is not only to derive classic seismotectonic fields over the Alpine belt as was done in previous studies with published focal solutions in the 3-5 Ml range (e.g. Delacou et al., 2004; Sue et al., 2007). Indeed only a few new earthquakes of this size occurred in the western Alps since then. Our goal is on the contrary to increase the spatial resolution of existing seismic deformation fields, by assessing which features appear robust based on a huge, although possibly noisy, set of small to moderate magnitude focal solutions. This goal is motivated by the increasing number of available seismic records in low deforming areas thanks to the enhancement of seismic networks in the recent years. In order to make the best use of these emerging datasets, we here selected 2215 focal solutions over many more that could be computed. We detail in the following why this unconventional focal mechanisms dataset, combined with state-of-the-art analysis tools, allows us to establish unsuspected spatial variations in the seismic deformation pattern in the Western Alps.

2) discussion:

We again thank reviewer 2 for his suggestions which helped us to improve the geodynamic discussion of our manuscript. As detailed below, we now address in more detail the issue of where our results stand regarding the ongoing debate on the state of the European slab beneath the Western Alps as well as the robustness and interpretation of compression.

Specific comments:

1) *Quality of FM-Quality: I have serious concerns related to the quality-control of the presented FM catalogue. About 1258 solutions (out of the 2215 total FMs) have ML magnitudes <2.0. In my experience, working on FMs in the Alpine region, unique, high-quality FM solutions in this magnitude range constrained from P-polarities alone are extremely rare. In almost all cases the solutions are highly uncertain and ambiguous and -if at all- only possible in regions of extremely dense station coverage. In addition, polarities in standard, routinely picked bulletin data (as used in this study) are full of mistakes/blunders and not very reliable unless reviewed by experienced seismologists with the goal to derive a reliable focal mechanism. This is likely even worse when data is combined from different bulletins. In combination, I expect severe uncertainties and ambiguities in the derived FMs. In less dense instrumented areas, I would*

*even consider solutions in the ML range 2.0-3.0 (769 FMs) as highly uncertain (at least partly), e.g. if the focal depth is poorly constrained and therefore take-off angles can be highly uncertain. In the provided FM catalogue (supplementary material) 94% of the FM solutions are of lowest quality class "D" (2076 FMs) indicating that the majority of the solutions is highly uncertain. The authors associate "D" qualities with "strike uncertainties" of 45-55 deg (and I assume it's even higher for dip and rake). It seems, however, that the a priori quality of the solutions is not taken into account in any of the applied inversion schemes. The uncertainties derived by the Bayesian inversion of P and T plunges in Fig. 8, on the other hand, are surprisingly small (<10 deg for the majority of the region, values I would associate with high-quality FM solutions). This uncertainty in P and T plunge seems therefore unrealistically small to me given the potential quality of the majority of the used "D" FM solutions and raises the question how much of the "small-scale variations" in the deformation regime is related to noise introduced by the large amount of low-quality FM solutions. The authors should extend their discussion on the potential uncertainties of the low-quality solutions and how it potentially affects their inversion results. E.g. they could use a priori weighting based on the solution qualities in their inversion and compare results against solutions without quality weighting. Also, in some of the maps, the locations of higher-quality solutions could be marked by circles of different colour to identify the regions which are constrained by more reliable solutions. In addition, I also miss a benchmark of the "automatically" FM solutions derived by the authors against existing, manually reviewed, high-quality FM or moment-tensor solutions published by French, Italian and Swiss agencies. This would help to assess the potential uncertainties and reliability of this catalogue.*

Concerning the quality of the dataset, first of all, as stated by the reviewer, we were only able to derive well-constrained fault plane solutions for small magnitude events (Ml < 2) thanks to the high quality of the seismic records. The records indeed result from a region of extremely dense seismic station coverage as mentioned by the reviewer. The relocation of the events in a 3D crustal velocity model provided tight constraints on the depth of the seismic events and thus on the take-off angles (Potin, 2016) used to compute focal mechanisms. Moreover the polarity dataset used in the present study was manually revised each year by Sismalp network seismologists. We would also like to stress here again that the quality flag is not a standard deviation provided on focal mechanisms but a summary of several uncertainty indicators. It thus only suggests that fault plane solutions may not be as constrained as A-C quality solutions, and by no means that these solutions are necessarily unreliable. To assess which features are robust within this qualitatively heterogenous fault plane solutions, we thus exploit the redundancy provided by the corresponding huge and spatially dense dataset by implementing a Bayesian surface reconstruction inversion scheme. As explained in section 2.4, we do weight focal mechanisms differently according to their a priori quality, and decipher, given the level of noise necessary to fit the data, how much of these uncertainties are needed to explain the input dataset.

Concerning the 1-sigma uncertainties provided by the Bayesian inversion which may appear low in comparison to the quality of the input data, this legitime question was clarified by adding the following paragraph in the revised version : "**As in any inverse problem, the level of**

uncertainty associated to the solution can only be interpreted relative to the level of resolution. Here, it is important to keep in mind that the solution model is smooth and represents an average over a given wavelength. The model uncertainties shown in Figure 8 are associated with this spatial average, and can therefore be much smaller than the data variability or data uncertainties (see Choblet et al, 2014 for more details)."

*2) Seismic strain rates: As the authors stated themselves, the seismic moment is dominated by the largest events in a region. I am therefore wondering how much the seismic strain rates in Figure 4 and the seismic "flux" in Fig. 7a are controlled by the largest events in this region. I would therefore suggest to plot the corresponding beachballs of events with ML>=4.0 in the background of Figure 4 and indicate the corresponding magnitude of these events. This might help the reader to understand if the corresponding strain regime and strain rate in the corresponding zonation is mainly dominated by the largest event within this zone. Is the relatively smaller moment release in the NE corner of Figure 7a real or just an artefact of the completeness of the author's catalogue? In Figure 2c, I see several M3-4 events in this region and I would have assumed to see a corresponding relative increase in moment release in this part as well.*

In the Kostrov summation which is illustrated in Figure 4, the computed strain rates indeed depend mostly on the higher magnitude events of each area, as mentioned in the main text (l.296). We acknowledge that this graphical information could be valuable to the reader and we plotted the corresponding features in Figure 4 as suggested. Regarding the low amount of moment release in the specific area NE of our network, we confirm that this represents an artefact due to a border effect, based on Figure 1 (less dense station coverage and lack of stations east of the area resulting in less events recorded than further west), and based on Figure S2, which shows the raw seismic moment prior to smoothing. We thus apply an additional mask on Figure 7 based on the density of seismic events.

*3) Discussion/Interpretation: In my opinion, the discussion part could be extended and improved. The discussion is a bit vague and general. Some aspects which could be extended:*
*● Role of slab dynamics in the Western Alps: Different models have been proposed on the slab structure: detached (e.g. Lippitsch et al 2003) vs. attached (e.g. Zhao et al 2016). In addition, if attached, rollback and delamination might play a role (e.g. Sue et al 1999). How do the latest insights into slab-models fit/support your observations? Do the presented results add new constraints to this ongoing discussion?*

We indeed omitted to mention that several tomography models over the western Alps suggest a slab break-off, while the recent study of Zhao et al. (2016) represents a possible evidence for a continuous slab down to a depth of ~ 300 km. We thus added the following paragraph to the discussion l.645 :
**"The current state of the European slab beneath the Western Alps remains to this day a matter of debate. Indeed, evidence for both a detached slab (e.g. Lippitsch et al., 2003,**

**Kissling et al., 2006, Diehl et al., 2009, Kästle et al., 2018) or a continuous slab (e.g. Piromallo and faccenna, 2004; Zhao et al., 2015; 2016) are claimed between different tomography models. The focal mechanisms derived in this study thus only provide seismic deformation styles up to 35 km, when the continuation of the European Moho beneath the Adriatic one, deeper than ~ 60 km, is under ongoing discussions. While our results constrain a depth too shallow to help decipher between these two end-member models, we rely on the literature to suggest that a recently (< 5 My, e.g. Lippitsch et al., 2003) detached slab, with its detached extent nowadays located beneath the eastern margin of the Western Alps, could induce extension as well as uplift, due to lithospheric rebound processes and/or mantle upwelling related to the sinking into the asthenosphere of more dense lithospheric material (e.g. Sternai et al., 2019). Slab dynamics can indeed cause transient dynamic topography (e.g., Faccenna and Becker, 2020), or influence exhumation processes (Baran et al., 2014; Fox et al., 2015)**."

As for the role of rollback and delamination, they may indeed introduce additional spatial variations of the vertical and horizontal deformation fields if attested. However, our results cannot bring any new constraints on this specific aspect due to the spatial scale and related depth resolution of our study, and we propose here a first order kinematic model that may be enhanced with more complex features in the future.

- *I do not understand how robust (and how to interpret) the N-S compression in the western Po plain shown in Figure 6 is. In Figure 5 this compression seems more SW-NE? and others (Delacou et al 2004, Eva et al 2020 (tectonics)) show also more SW-NE or even E-W directed compression. Is this due to very few mechanisms of one cluster? How well are they constrained? Why did the former compressional domains east and west of the Arc (e.g. Delacou et al 2004) "disappear/reduce" in your results? Is it possible that the weight of the compressional events is just suppressed by the larger number of additional small (and likely noisy) low-quality FM solutions of oblique type? Or did the former reverse type mechanisms change to strike-slip in your FM calculation? If so, are the new solutions better constrained then the previous solutions?*

The robustness of the various compressive patterns is discussed in section 4.2 (l601-611). As shown in section 3.3, the occurrence of compression in the Po plain is a robust feature. However the probabilistic approach used here allows us to decipher robust deformation modes throughout the area and not to assess the robustness of the corresponding azimuthal directions. It appears from section 3.2 (Figure 5 and table 2) that the azimuthal direction of compression in the Po plain is less robust than for the surrounding areas, and varies from N-S to NE-SW between the different inversions. Despite this uncertainty, our results suggest compression with azimuth ranging from ~10° to 45° CWN and differ from some other studies, which identify either E-W compression (Delacou et al., 2004) or N-S compression (Eva et al., 2020). The depth of the corresponding seismic events also varies significantly in the literature. Thus the interpretation of this localized compressive pattern is still a matter of debate, and the spatial resolution of the present study, which aims at studying deformation at the scale of the Western Alpine arc, does not allow us to settle it.

As for the compressive pattern previously observed along the Belledonne fault (e.g. Delacou, 2004) it appeared controlled by very few compressive mechanisms, which were not well resolved. Our study adds both strike-slip and compressive mechanisms in this area. However the few compressive solutions are less well constrained than the strike-slip ones, which explains that compression almost disappears in the statistical approach.

Lastly, the compression south-west of the Alpine arc disappears from the inversion results due to the higher number of equally robust strike-slip mechanisms compared to reverse ones (Figure 3).

● *How can the contrasting juxtaposition of compressive and extensional domains at the eastern boundary be explained (e.g. Figure 9c)? Similarly, can the mix of strikeslip and normal mechanisms within the core of the Western Alps be simply explained by a general transtensive regime (some faults take up strike-slip, others the normal component)? Are principal axis compatible with transtension (i.e. is the orientation of the T-axis consistent and only B and P flip?). If not, it is difficult to explain with a homogeneous far-field stress. A final sketch figure indicating the overall kinematics would certainly help to summarize and better explain the proposed model of vertical deformation controlling the patches of extension while rotating Adria controls the over-all strike-slip regime. Still the question remains why there is little spatial correlation between "extensional patches" and surface uplift. Also, you should compare your re-sults and model in more detail with the recent paper of Eva et al 2020 (Tectonics), who propose a similar 3D seismotectonic model of the same region.*

We added the following paragraph to the discussion: "**The contrasting juxtaposition of extension and compression occurs in a region of complex geometry and complex processes related to the Alpine orogeny (plate boundaries, Ivrea body). It has sometimes been interpreted in the literature as a border effect of gravitational collapse of the Western Alpine chain (e.g. Delacou, 2005), as well as a marker of indentation resulting from Africa-Eurasia current plate dynamics in other studies (e.g. Eva et al., 2020). In both cases, sharp variations in the stress field are expected. In the case of the gravitational collapse for instance, stresses depend on the spatial derivative of the load, and to some extent varies like the derivative of the crustal thickness and topography: clearly, these variations occur on short spatial scales. While we cannot decipher which processes are at the origin of this very specific and local pattern, we stress that too few compressive patterns are retrieved at the border of the chain to support the gravitational collapse scheme, and that indentation by Adria or Corso-Sardinia blocks does not appear to be the main process controlling crustal deformation nowadays in the Western Alps, since a majority of extension and strike-slip mechanisms are found, drawing a self-consistent transcurrent deformation field over the whole Western Alpine region.**"

As for the occurence of both extension and strike-slip, these features set up the main finding of our paper as stated at the end of the reviewer's general comment. We indeed explain this particular transcurrent pattern as the result of the interaction between (i) far-field forces related to the present-day rotation of Adria wrt stable Eurasia, which results in the homogeneous strike-slip field observed at the regional scale, and (ii) intrinsic forces possibly related to isostatic

processes and slab dynamics, which can result in the small scale spatial variations of extension that are retrieved in the core of the belt.

Last, we do agree that a final sketch will make our point clearer. We added in the revised version of the manuscript a 3D sketch to the discussion section, summarizing our main findings and how they could relate to the main processes possibly involved in crustal deformation in the Western Alps. Regarding the spatial decorrelation which is evidenced in our work as in other previous studies between maximum of deformation (i.e extension, whether geodetic horizontal component or seismic deformation) and vertical uplift measured from geodesy, we suggest that extension, which is imaged both by geodesy and by seismotectonics with a maximum in the South-Western Alps, could be generated by the same processes involving uplift, even if it appears spatially decorrelated from uplift maximum. The aforementioned processes (isostatic adjustments and possible slab detachment) may thus infer crustal deformation which is released both seismically, through extension, and aseismically, through uplift. Seismic deformation would thus locates preferentially in the inner zones, along the two main seismic arcs inherited from the complex orogen history of the Alps, while aseismic deformation locates preferentially on the North-Western part of the Alps, possibly due to additional short wavelength processes such as differential erosion and/or differential ice mass loss.

4) The English is reasonable; however, I spotted some smaller issues I have listed in the following list. I would recommend another round of proofreading by the authors. Detailed Comments:

We took the suggested corrections into account for the revised version and answer below to the various interrogations of the reviewer:

*- L. 80: Indicate the exact period of the catalogue: from 19XX to 20XX. What is the estimated Mc of this catalogue? All events included from all contributing agencies or only above a certain magnitude?*

We added the exact period of the catalogue which is from 1989 to 2013. The Mc was estimated to 1.15 for the whole Western Alpine region based on a Gutenberg-Richter analysis of the complete dataset corresponding to the map in Figure 1 .

*- L. 119: What is the S-phase needed for? Are you using S-polarities as well? As pointed out above, the azimuthal/take-off gap criteria are not sufficient to exclude grossly ambiguous FM solutions. I would have rather considered HASH's number of FM families (use only solutions with 1 family of solutions -> +/- unique solution) and number of possible solutions as quality criteria.*

The minimum number of one S wave criteria is applied in order to ensure to select well located events, especially in terms of depth constraints. Amongst the computed focal mechanisms, only 4 have multiples with two possible solutions. These events are marked with a star flag (*) in the revised appendix of the manuscript.

*- L. 126: "uncertainties on picking" -> shouldn't it be "in polarities"? Did you use quality weights for polarities (e.g. U/D vs +/-) for the grid search? Did you use the uncertainty in TOA? How is this uncertainty estimated and how is it implemented in your HASH Procedure?*

We indeed use two levels of polarity uncertainty (U/D and +/-) in the focal mechanisms inversion. We did not use any weighting for take-off angles since localisations, take-off angles

and azimuths were computed in the same 3D velocity model and are therefore all of the same order.

*- L. 127: Was ML recomputed by Potin or is this basically the original SISMALP ML? Is ML really always measured from S or is it simply the peak amplitude on the horizontal components (regardless of P,PmP, S, SmS, surface wave)?*

Ml have been recomputed based on events relocation. However the bias that would have been introduced if it had not been done would be small (average of the location differences is 5 km, see Potin, 2016) compared to the original uncertainties on the magnitudes. The computed Ml is defined by Richter (1936), and is indeed simply the peak amplitude but on all three components, since for some shallow events amplitude is maximum on the vertical component.

*- L. 144: "The representation of style . . ." Rephrase this sentence.*

We rephrased it as follows: "**One can use ternary diagrams to graphically represent intermediate faulting styles between pure strike-slip, pure normal or pure reverse motion.**"

*- L. 180: This is a bit confusing, first the authors summarize different scaling-relationships to convert Ml to Mw, referring to a study proposing a polynomial fit but then it seems the authors use a 1:1 relationship without too much explanation why this is valid. What is the mistake in term of Mo of this simplification for larger events (M>4.0)? Why not using the already established relationship?*

The recent work by Laurrendeau et al. (2019) has been demonstrating that already established relationships (Cara et al., 2015) are estimated at the national scale and do not take into account regional variations in crustal attenuation characteristics. This work also demonstrates that events in south-eastern France, based on events available in both Ml and Mw, are closer to a 1-1 relationship than to previously established relationships (Cara et al., 2015) which tend in this region to underestimate Mw of events with Ml > 3.0.

*-L. 200: In classic stress-inversion the slip is assumed to be on the active plane (not on the auxiliary one). Others (e.g. Kastrup et al 2004) therefore down-weighted solutions for which the active plane is unknown (e.g. no information from relative relocation available). How did the authors address this problem in their stress-inversion strategy?*

We choose not to apply any weighting for fault planes, following previous studies in this region (e.g. Delacou et al., 2004) since we work at the regional scale with small magnitude earthquakes for which faults are often not identified.

*-L. 243: The methods "enables us to assess whether formal uncertainties on fault planes . . . are over or underdetermined". What is the result of this assessment then for your data? What are the formal errors of your focal mechanisms?*

We  took into account the fault plane uncertainty as input error on our focal mechanisms. **We were able to assess through the Bayesian inversions that these formal errors are overestimated by a factor of ~ 0.6. This is now specified in the revised version of the text.**

*- Figure 5: I see some differences in P and T between the two inversion methods which seem larger than the* 1sigma *estimated by the Bayesian inversion. Any explanation for these differences?*

The stress inversion in the seismotectonic zonations is a different procedure than the Bayesian interpolation of P and T angles. It is therefore difficult to directly compare results and uncertainties associated with the two different approaches.

*- Line 344: You mean both methods give similar results for the orientation of sigma3? Consider rephrasing this sentence.*

Indeed. We rephrased it to "**The least compressive stress axis (σ3) presents quite similar azimuth values [...] except [...]**".

*- L. 381: Not sure I understand this sentence. What are "longitudinal directions/faults"? Isn't the mentioned fault striking NE-SW?*

It meant the strike of the belt versus the strike of the fault indeed. We rephrased it to "[..] **with a rotating state of stress compatible with dextral motions on faults parallel to the strike of the belt (such as the Belledonne fault, Thouvenot et al., 2003)** ".

*- Figure 7b) why not add a colour bar (similar to Delacou et al 2004) rather than describing the meaning of the colours in the caption?*

As explained section 2.2 (l.153), a focal mechanism cannot be reduced to only 1 dimension. While Delacou et al. (2004) simplified the style of deformation to one dimension, we consider here, in addition to purely compressive or extensive, also purely strike-slip as well as intermediate transtension and transpression motions, thus requiring at least two axes to describe the focal mechanism. We thus refer to the ternary diagram for the meaning of the 2D color code.

*- L. 459: This extensional zone at 10 km: How well is the depth constrained for the associated events? This zone north of the Valais is more complicated than strike-slip. It contains all kinds of mechanisms: Strike-slip, oblique normal, oblique reverse, most likely it consists of an array of strike slip faults connected by releasing and restraining bends/step-overs. See e.g. Diehl et al. 2018, Earthquakes in Switzerland and surrounding regions during 2015 and 2016, https://doi.org/10.1007/s00015-017-0295-y. A bit north of the major strike-slip zone, towards the Alpine front, there is indeed a zone of extensional events (e.g. M4.3 Chateau-D'Oex earthquake of 2017, Jaun M3.8 event of 1999), which is maybe what the authors image in their cross-section 1 (needs to be checked). It is described and discussed in a recent publication which is currently in press and should be published online soon (Diehl et al. 2018, Earthquakes in Switzerland and surrounding regions during 2017 and 2018, https://doi.org/10.1186/s00015-020-00382-2). However, this extensional domain is much shallower than in the author's cross-section (uppermost crystalline basement or Mesozoic sediments, likely <5 km).*

The depth north of the Swiss Valais is as well constrained as in any other part of the crustal velocity model covering the Western Alps, thanks to a high density of swiss stations and to the high quality of related seismic records. While depth uncertainty is a research topic by itself, we remind here that the locations in the 3D velocity model are constrained by both P and S waves and differ, on average on the whole area, of about 5 km (in 3D) from the locations previously estimated by Sismalp in a 1D velocity model (Potin, 2016). This specific area is however complicated as stated in the above comment and we now refer to Diehl et al., 2018; 2021 in addition to Maurer et al (1997) and Eva et al (1998) to further help readers specifically interested in these local scale features.

*- Figure 9: It would be helpful to add additional geological reference information from*

*geological profiles, like position Alpine front, Ivrea body, etc. What are the tiny black dots (difficult to see)? Projected earthquakes (everything) or just earthquakes with corresponding focal mechanisms used in the inversion/regression? I would make the symbols corresponding to FMs bigger (maybe as circles), try colour code the quality of the mechanisms. This would help to distinguish parts well constrained by data from areas with inter- or extrapolated values.*

The dots indeed refer to the focal mechanisms used in the inversion in order to highlight areas well-constrained by data. We made the corresponding symbols bigger. We refer to Figures S3 and S4 for robust versus extrapolated values. The location of geological massifs are indicated by acronyms, while the location of geophysical characteristics are indicated on the tomography models on Figure 10.

- Figure 7b/text around line 517: Why not show Figure 7b for different depth intervals (similar to tomographic results) rather than projecting everything to one layer? This would lead to a "patchier" distribution with more white-spaces, but would avoid some of the misunderstanding due to vertical projection?

The inversions were indeed also performed for different depth intervals. The corresponding interpolated deformation fields (mean of the probabilistic distribution) were added to the supplementary material to help the reader decipher which mechanisms, at which depth, most constrain the surface-projected reconstruction of the deformation field shown in Figure 7b.

- L. 540: ". . .follow the structure of the European crust. . ." Not sure in terms of what? You mean in terms of dip? Or lithology? Should be more specific.

**"... follow the structure of the European crust in terms of dip".**

- L. 543: "former slab" What do the authors mean here? Does this "former slab" relate to the possibly detached slab? As mentioned above, this discussion needs to be Extended.

Yes it does. This point is now enhanced in the discussion as developed in the answer to specific comment 3) above.

- Figure 10: Are the beachballs shown on the profile in b) lower hemisphere projections (as in map view) or cut along the profile (projections)? Since they all plot on each other it's difficult to see anything. . . In caption of 10b, why not simply say: Dashed lines represent the European and Adriatic Moho after (???). Moho in 10b is from Spada as Well?

The focal mechanisms in all cross-sections are represented as cut along the profile. Ok for caption 10b : "Dashed lines represent the European and Adriatic Moho after (Solarino et al., 2018). "

- L 586: ". . . movement along the longitudinal Alpine strike" Not sure what the authors mean here.

replaced by "dextral motion along the strike of the belt."

*- L. 650: "low noise transcurrent motion" Low-noise in terms of what?*

We rephrased it to "robust transcurrent motion"

*- L. 655: "but may also add another component" -> What is this other component?*

The Adriatic plate rotation is suggested to add a strike-slip component to the observed deformation field. We rephrased it to "The counterclockwise rotation of Adria with respect to

stable Europe (e.g. Calais et al., 2002; Serpelloni et al., 2005, 2007) largely counterbalance buoyancy forces (Delacou et al., 2005)".

*- L. 655: "While a purely plate-related geodynamic model seems discarded by now. . . our observations may revive the role of plate motion. . ." This sentence doesn't make much sense to me. Is it discarded or not? What other process should explain seismicity and deformation?*

As explained in the discussion section, the processes driving seismicity and deformation in low deforming areas, and especially within the Western Alps, is a long ongoing debate. While some studies (e.g. Calais et al., 2002) suggested that crustal deformation could be related to the sole active plate-tectonic processes, other studies suggested that surface- or slab- related isostatic processes could also explain some crustal deformation and seismicity (e.g. Sternai et al., 2019, Mazzotti et al., 2020, Salimbeni et al., 2018, Eva et al., 2020). The main finding of our study, as summarized in the general comment of this review, is that both active plate tectonics and other neotectonic processes must be involved to explain the observed deformation field. We made this point clearer with the final sketch and the corresponding discussion in the revised version.

*- L 695: Why not add used polarities and take-off/azimuth angles to the supplementary material to allow others to assess the quality of mechanisms.*

The relocalized catalog will be the object of a specific publication, to which we will refer in the final version if available.

*- L 698: What about all the data added from other networks? Nowadays most networks have DOIs and should be cited with their corresponding network code and DOIs.*

We added the DOIs to the citation of the national networks when available.

**References:**

Amelung, F., & King, G. (1997). Large-scale tectonic deformation inferred from small earthquakes. *Nature*, *386*(6626), 702-705.

Ammirati, J. B., Vargas, G., Rebolledo, S., Abrahami, R., Potin, B., Leyton, F., & Ruiz, S. (2019). The crustal seismicity of the western Andean thrust (central Chile, 33°–34° S): Implications for regional tectonics and seismic hazard in the Santiago area. *Bulletin of the Seismological Society of America*, *109*(5), 1985-1999.

Bonjer, K. P. (1997). Seismicity pattern and style of seismic faulting at the eastern border fault of the southern Rhine Graben. *Tectonophysics*, *275*(1-3), 41-69.

Courboulex, F., Larroque, C., Deschamps, A., Kohrs-Sansorny, C., Gélis, C., Got, J. L., ... & Mondielli, P. (2007). Seismic hazard on the French Riviera: observations, interpretations and simulations. *Geophysical Journal International*, *170*(1), 387-400.

Diehl, T., Husen, S., Kissling, E., & Deichmann, N. (2009). High-resolution 3-DP-wave model of the Alpine crust. *Geophysical Journal International*, *179*(2), 1133-1147.

Diehl, T., Clinton, J., Deichmann, N., Cauzzi, C., Kästli, P., Kraft, T., ... & Wiemer, S. (2018). Earthquakes in Switzerland and surrounding regions during 2015 and 2016. *Swiss Journal of Geosciences*, *111*(1), 221-244.

Diehl, T., Clinton, J., Cauzzi, C., Kraft, T., Kästli, P., Deichmann, N., ... & Wiemer, S. (2021). Earthquakes in Switzerland and surrounding regions during 2017 and 2018. *Swiss Journal of Geosciences*, *114*(1), 1-29.

Kissling, E., Schmid, S. M., Lippitsch, R., Ansorge, J., & Fügenschuh, B. (2006). Lithosphere structure and tectonic evolution of the Alpine arc: new evidence from high-resolution teleseismic tomography. *Geological Society, London, Memoirs*, *32*(1), 129-145.

Maurer, H. (1993). *Seismotectonics and upper crustal structure in the western Swiss Alps* (Doctoral dissertation, ETH Zurich).

Piromallo, C., & Faccenna, C. (2004). How deep can we find the traces of Alpine subduction?. *Geophysical Research Letters*, *31*(6).

Salimbeni, S., Malusà, M. G., Zhao, L., Guillot, S., Pondrelli, S., Margheriti, L., ... & Zhu, R. (2018). Active and fossil mantle flows in the western Alpine region unravelled by seismic anisotropy analysis and high-resolution P wave tomography. *Tectonophysics*, *731*, 35-47.
Sue, C., Thouvenot, F., Fréchet, J., & Tricart, P. (1999). Widespread extension in the core of the western Alps revealed by earthquake analysis. *Journal of Geophysical Research: Solid Earth*, *104*(B11), 25611-25622.

Thouvenot, F., Fréchet, J., Tapponnier, P., Thomas, J. C., Le Brun, B., Ménard, G., ... & Hatzfeld, D. (1998). The ML 5.3 Epagny (French Alps) earthquake of 1996 July 15: a long-awaited event on the Vuache Fault. *Geophysical Journal International*, *135*(3), 876-892.

Thouvenot, F., & Bouchon, M. (2008). What is the Lowest Magnitude Threshold at Which an Earthquake can be Felt or Heard, or Objects Thrown into the Air?. In *Historical Seismology* (pp. 313-326). Springer, Dordrecht.

Zhao, L., Paul, A., Malusà, M. G., Xu, X., Zheng, T., Solarino, S., ... & Zhu, R. (2016). Continuity of the Alpine slab unraveled by high-resolution P wave tomography. *Journal of Geophysical Research: Solid Earth*, *121*(12), 8720-8737.

---

## Author Response (AR2)

Report #1

I have read the revised version of Mathey et al's manuscript as well as their response to the comments raised by the reviewers related to the previous version. All references to lines and pages refer to the version including the track changes.

A) The authors responded at length to the main concern related to the quality of the computed focal mechanisms (FM) raised by reviewer #1 and myself. I can follow their arguments partly, however, I am still not entirely convinced of the value and the reliability of the FMs calculated for magnitudes <2.5. This is mainly because it largely contradicts my daily experience in earthquake analysis in the Alps, in which reliable mechanisms for magnitudes <=2.5, even with comparable dense (or even denser) networks, are rather exceptional. If possible at all, it requires extremely careful manual review during picking as well as the FM computation.

I am quite surprised to read that, according to the author's response, only four of their entire 2200 FMs (about 50% have MI <2.0) have more than two possible solutions. Again, this completely contradicts my own experience, maybe this is a misunderstanding of what I had in mind. In my own HASH implementation, I use the following parameters, which proved to result in very realistic uncertainties:

dang = 2.0 ! minimum grid spacing (degrees)
maxout = 500 ! max number of acceptable mechanisms output
cangle = 30.0 ! mechanisms are "close" if less than this angle apart (degrees)
prob_max = 0.1 ! probability threshold (cut-off) for multiples (e.g., 0.1)

Maybe the authors use less strict parameters (especially for cangle) or allow for a less broad solution space? Again, in my experience the majority of mechanism result in multiple-solution families for M<2.0 events. The authors also argue that calculating FMs for such small events is mainly possible because of the extremely dense network. However, the network used in their study shown in Fig. 1 is not "extremely" dense on average in my opinion. Unfortunately, no km scale is provided in this map (as in most others), but in many regions the average spacing looks more like 30-50 km to me (which I would not call extremely dense nowadays). In addition, some stations (I guess the lines) are from temporary experiments. One more aspect: Since the authors report the average of all acceptable solutions of HASH, I am wondering if this averaging might lead to a systematic bias towards strike-slip mechanisms in case of extremely poorly constrained solutions. I apologize if this appears extremely pedantic, but my point here is to avoid that readers less experienced with FM calculations get overly confident when using this FM-catalog in future studies. The quality classification currently used seems not sufficient to me to really distinguish between reliable and not reliable solutions and I would have preferred to simply use e.g. the variation in the "acceptable" solutions returned by HASH as a quality measure rather than the predefined classes based on gap etc. Depending on the type of mechanism, the source depth and the distribution of polarities on the stereonet, such parameters can be completely underestimating the true uncertainty of the FM. Using HASH parameters as listed above, in comparison with some manual revised reference-solutions, should have provided better and more robust quality classifications as the one currently used.

Nevertheless, I admit that the overall results look reasonable and consistent with previous studies and maybe the FM qualities impact the overall results less than expected. Therefore, as practical solutions to make the quality classification of the presented FMs more transparent for readers I suggest to:

1) Provide a table in the supplement with all the HASH inversion parameters used by the authors to make the results reproducible.

2) As nowadays commonly required in many journals, provide the full dataset used to calculate the mechanisms e.g. in a data repository. The authors disapproved that suggestion according to their response because data seem to be used for another unpublished study. But providing only FM-basic information like Stationcode, polarity, polarity-quality, azimuth-angle, take-off-angle and basic hypocenter information (OriginTime, LAT, LON, Depth, Mag) should be sufficient (no information on phase picks is required if that is still used for something else). With this basic FM-data provided, everybody could use her/his own tools to assess the corresponding FM quality. I strongly encourage the authors to make that basic FM-data available alongside with their publication.

3) With HASH parameters similar to the ones listed above generate a figure which shows: a) number of "accepted" FM solutions (all strike-dip-rake combinations which fit the data within their uncertainties, which means allowing for at least 1-2 outliers, HASH parameter) vs. magnitude for each event. b) similarly, plot number of solution-families (make sure cangle, prob_max are reasonable) vs. magnitude for each event. I would expect that the number of solutions & solution-families reduce for larger magnitudes. Such figure would provide additional information on the confidence of the FM solutions of low magnitude events.

→ We thank the anonymous reviewer for his continued help improving our manuscript. While we stress that the described seismic networks allow us to compute some reliable small magnitude focal mechanisms, we acknowledge that the reader should be well aware that the computed FM dataset is suitable for our large regional scale study, but may not be suitable for any local scale study due to the heterogeneity in the quality of the FM computed in the present study. Thus we followed all the suggestions to help the reader understand our variable quality focal mechanism huge dataset. We indeed define mechanisms "close" together as those having a difference angle < 45°, and choose to ignore multiples with a low probability by setting the probability threshold for multiples to 10%, which explains the low number of multiples even for low magnitude events in our dataset. As for the use of the average of all acceptable solutions, this output is indeed the one used by the software's authors (Martínez-Garzón, 2014), for which they do not mention any potential bias in HASH FM computation for less constrained solutions.

1) We now provide in the supplement a table (Table S1) with the control parameters used for HASH focal mechanisms computation.

2) We now provide in the supplement the dataset ("computedFM_HASHformat.inp") corresponding to the 2215 computed focal mechanisms in HASH input format.

3) We added the suggested plots in the supplement (Figures S7 and S8) in order to help the reader assess the level of constraints on focal mechanisms depending on magnitude range.

B) While reading the manuscript a second time, I realized that section 3 (which seems the result section) extends over 14 pages (of 35 total). It seems to contain already a lot of interpretation and discussion and all the details listed in this section make it difficult to not lose focus. I would suggest to consider shortening this part to the essential findings. In addition, there are still issues with the English, some sentences are rather awkward or unclear. I listed some examples below, but I definitely recommend another iteration of thorough proofreading by the authors.

→ As suggested by the reviewer we removed some detailed description of the results that could be observed on the figures and stick to the major findings of the results section in the main text. We additionally made the English corrections suggested and went through an in-depth proofreading of the manuscript which improved its overall fluency.

C) Final comment: Isn't the fact that extension in the southern part appears to have no corresponding uplift signal in the geodetic data related to the depth of this extensional zone? In profile 4 extension seems >15 km, isn't this why no geodetic signal is seen at the surface? Therefore, the comparison of geodetic data and (surface-projected) seismotectonic results and derived conclusions in Fig 11 might be of limited value? Maybe a better comparison would have been (as proposed in my previous comments) to make a stress-regime map limited to FMs in the upper crust (e.g. <10 km) and compare that to the geodetic data?

→ As suggested in the previous review, we tested a stress-field map limited to the FM in the upper crust (Figure R2). However we did not add it to the manuscript since it appears consistent with the one derived using all FM with their complete depth range (Figure 6). Concerning the comparison of seismic and geodetic deformation, we stress that geodetic results also have their own uncertainties. The southern part of the study area, which shows limited uplift signal in the geodetic map from Sternai et al. (2019), is subject to more uncertainties than other parts since it relies on a limited number of young GNSS stations from the Italpos network. This area indeed appears affected by null to moderate uplift depending on the GNSS solution considered (e.g. Kreemer et al., 2020[1]).

[Figure]

Figure R2. Stress-field derived on a 0.5°x0.5° grid using focal mechanisms in the 0-10 km depth range only. The inversion is performed with MSATSI if at least ten focal mechanisms fall within each cell. The orientations of the deviatoric stress tensors appear compatible with the one retrieved using the complete depth range of focal mechanisms (Figure 6).

In addition to these general comments, I have specific ones listed below.

Detail comments:
- Line 19: "down to low magnitudes" -> rephrase. Give exact numbers instead
→ We rephrased it to "records in the 0-5 magnitude range" as stated l.84.
- Line 24: "since 1989" -> indicate entire period of this study: 1989-2013
→ ok.
- Line 33: "Compression is robustly…" -> Maybe: "Robust indications for compression are only observed at the boundary between the Adriatic and…"

→ ok.

- In many places: be sure it's correct: "short-wavelength" vs "short wavelength"

→ we corrected it to "short-wavelength" at lines 34, 40, 730 and 764

- L. 79 (caption Fig 1): -> "… can be clearly identified in the seismicity"

→ ok

- L. 88: "stress oriented inversion" -> "stress inversion"

→ ok

- L. 130: Why not simpler: "The preferred solution corresponds to the average solution of all possible acceptable solutions…"

→ ok

- L. 131: "The HASH code…"

→ ok

- L. 134: The statement on the Ml computation does not make sense in this location. Move it somewhere to line 185 where you talk about magnitudes. In addition, the statement still seems wrong, it's not the maximum S-wave, it's the maximum of P or S wave, right?

→ It is indeed computed as the maximum amplitude of the signal. We moved it to line 186 and rephrased it to "The local magnitude (Ml) of the catalogue is based on the maximum amplitude among all three components of the signal. A double conversion toward Mw has been proposed by Cara et al. (2015), through another national-scale local magnitude (Ml LDG)."

- L. 136: "Over the 2215" -> "From the 2215…"

→ ok

- L. 163: "… on the focal plane solution is lost"

→ ok

- Figure 2: The axes-annotations/labels are not readable in a,b,c. Make them bigger!

→ We made Figure 2 bigger so that labels are now readable.

- L. 220: The MATSI software is based on the method of Hardebeck and Michael 2006, no? In the MATSI paper, Vavrycuk 2014 is not mentioned at all. Please check this…

Figure 3: The axes-annotations/labels of the Kaverina-diagrams are not readable at all. Make them bigger! Also, the dots themselves are too small…

→ MSATSI is indeed based on SATSI from Hardebeck and Michael (2006), which makes use of the same inversion procedure (least-square inversion) as SI (Vavrycuk 2014), which was previously mentioned in the manuscript, as opposed to grid-search inversions such as FMSI (Gephart, 1990). To make it clearer we rephrased it to "We used the MSATSI (Matlab Spatial And Temporal Stress Inversion, Martínez‐Garzón et al., 2014) software to perform an inversion for each cell encompassing at least 10 focal mechanisms." We also modified Figure 3 to make it readable.

- L. 309: what do you mean with "time/energy relation"? You mean moment rate?

→ We modified the sentence as follows: "The short time span of the observations (24 years) prevents from thoroughly investigating seismic energy release spatial and temporal variations".

Figure 4: Caption: Explain what symbol delta refers to, I assume its dip?

→ delta indeed stands for the dip. We now refer to table 1 in the caption for symbol explanations.

- L. 357: "Lest" -> "Least"

→ corrected

- L. 381: Here you could also compare the results to others studies with similar results

→ We added the references to the following studies l.381 :

"The orientations of the strike-slip tensors appear in good agreement with previous studies, whether at the regional (e.g. Delacou et al., 2004) or at the local scale (e.g. Bauve et al., 2014; Kastrup et al., 2004). However the orientation of the extensive tensors appear slightly less perpendicular to the belt than previously observed (e.g. Sue et al., 1999; 2007b; see section 4)." As stated by the reviewer though, the result section is already quite extensive. For this reason a more detailed comparison of orientations with previous studies is found in the discussion section (section 4.2 l.610-645).

- L. 409/10: Isn't it "projected to" the surface?

→ ok

- L. 432: NW or NE Switzerland? Don't you mean NE?

→ corrected

- L. 443: consistent with… or according to… ?

→ ok

- Figure 8: Isn't the higher variation in P expected in a transtensional regime (where P and B are flipping while T remains the same)?

→ Figure 8 represents the uncertainty on P and T mean distributions. Higher uncertainties on P does not necessarily imply higher variations in P plunges. Indeed such variations can be well resolved while more homogeneous areas (i.e. P plunge constant, Fig.7) can be related to higher uncertainties, for example where the data is too scarce.

- Several places: "in the overall" -> "in general"

→ ok

- L. 486: Do you mean "consistent with the surface-projected results"?

→ Yes, corrected.

- L. 508: "narrow band of strike-slip deformation along…"

→ ok

- L. 529: "surrounded by a … regime, especially..."

→ ok

- L. 537: … by a slightly increasing… -> rephrase! By how much?

Rephrased : "It appears that extension is characterized by a depth increasing from ~10 to ~20 km from North to South"

- L. 540: "geological structures" what do you mean? Lithologies? Tectonic units? Specify

→ "[...] while cutting through external geological units both north and south of the profile."

- L. 543: "artefacts" -> caused by what?

→ These artefacts are likely introduced by the meshing of the model.

- L. 558: "of all mechanisms resulting from the projection to the surface"?

→ yes, modified accordingly.

- L. 560: "exemplified" -> "documented"

→ ok

- L. 576: "does not appear to be controlled by the geometry of the former European slab. Both extensive…" COMMENT: I still don't really understand what the authors mean here with the "former European slab"? All I see in this figure is the geometry of the Moho. And why "former"? isn't this still European crust/lithosphere? Rephrase this statement! Same later around line 581.

→ we referred to the fact that convergence appears to be over beneath the western Alps from geodesy and related kinematic models. We rephrased it for clarity :

"the distribution of the style of deformation does not appear controlled by the structure of the European Moho [...] thus coincides with the boundary between the European crust and the Adriatic one."

- Figure 10 caption. In the caption you should describe how the beachballs are projected (they are cut and projected right?) Mention the meaning of the colors of the mechanisms. Since the authors plot all of them, not much detail can be seen.

→ We added the following information: " a) Cross-section of computed focal mechanisms (vertical sections) projected (see width in c)) along ECORS-CROP seismic reflection profile, modified from Marchant and Stampfli (1997). Green: Strike-slip mechanisms; blue: normal mechanisms; red: reverse mechanisms. [...] b) Focal mechanism cross-section (vertical sections) projected along CIFALPS profile (projection width in c)) of local earthquake tomography, modified from Solarino et al. (2018), color-coded as in a)."

- L. 602: repetition…

→ removed

- L. 639: -> "… extension previously proposed (CITATIONS)."

→ We added the following citations : (Eva et al., 1997; Delacou et al., 2004; Sue et al., 1999; Sue et al., 2007b).

- L. 660: I have to admit that I still don't really understand how such "gravitational collapse" causes N-S directed compression in the Ivrea mantle. Try to rephrase this statement.

→ We do agree. Our main point is that gravitational collapse cannot explain the asymmetrical compression observed at the periphery of the belt. We rephrased it for clarity : "While we cannot decipher which processes are at the origin of this very specific and local pattern, gravitational collapse schemes fail to explain the localized compressive pattern observed at the border of the chain, and indentation by [...]"

- L. 679: "collocate" -> "correlate"?

→ ok

- L. 681: As the authors mention earlier, the extension in the south is much deeper (profile 4). Isn't this the reason why the signal in the geodetic data (at surface) is missing? Where the authors see correlation (S-Valais), the seismotectonic extension is shallow…

→ As explained in the answer to comment C), the vertical motion derived from geodesy is poorly resolved in the south-eastern part of the area. If some extensive and uplifting areas appear spatially correlated, we here only stress that maximums of respective deformations appear disconnected.

- L. 699 and elsewhere: Try to avoid the use of "Indeed,"

→ ok

- L. 703: "When the continuation…" This sentence should be rephrased, bit confusing. Maybe what the authors mean: "…, while the debated continuation … is likely deeper than about 60 km."?

→ yes, modified accordingly.

- L. 715: "can't" -> "cannot"

→ ok

- L. 720: Still I don't fully understand what the authors mean with "purely plate-related geodynamic model" vs their "role of plate motion". Isn't everything plate-tectonic related? The authors should improve this part of the discussion.

→ By "purely plate-related geodynamic model" we mean without involving interactions with intrinsic surface (erosion and deglaciation) and deep (mantle- and slab-related) processes in order to explain present-day seismicity and surface deformation; while by "the role of plate motion" we mean in comparison with the role of surface and deep processes which have also been proposed as being at the origin of both crustal deformation and seismicity (e.g. Mazzotti et al., 2020; Calais 2016[2]). We added these following information in the discussion:

"While a purely plate-related geodynamic model seems discarded by now (D'Agostino et al., 2008; Devoti et al., 2008) due to the evidence of both extension and uplift in many places all along the Western Alpine arc, which cannot be explained by plate kinematics alone, we stress that our observations revive the role of plate motion in interaction with buoyancy forces in an attempt to explain the current Alpine kinematics and seismicity (Figure 12)."

- Figure 12 caption: -> "… block diagram"

→ Ok

- L. 743: "resolution without the use of a priori … zonation"?

→ ok

References:

[1]Kreemer, C., Blewitt, G., & Davis, P. M. (2020). Geodetic evidence for a buoyant mantle plume beneath the Eifel volcanic area, NW Europe. *Geophysical Journal International*, *222*(2), 1316-1332.

[2]Calais, E., Camelbeeck, T., Stein, S., Liu, M., & Craig, T. J. (2016). A new paradigm for large earthquakes in stable continental plate interiors. *Geophysical Research Letters*, *43*(20), 10-621.